# Anchor extension: a structure-guided approach to design cyclic peptides targeting enzyme active sites

Parisa Hosseinzadeh [1,6], Paris R. Watson [2,11], Timothy W. Craven[1,11], Xinting Li[1,11], Stephen Rettie[1,3,11], Fátima Pardo-Avila[4], Asim K. Bera[1], Vikram Khipple Mulligan[1,7], Peilong Lu [1,8], Alexander S. Ford[1], Brian D. Weitzner [1,9], Lance J. Stewart [1], Adam P. Moyer [1,5], Maddalena Di Piazza[1], Joshua G. Whalen[1], Per Jr. Greisen[1,10], David W. Christianson [2] & David Baker [1✉]

Despite recent success in computational design of structured cyclic peptides, de novo design of cyclic peptides that bind to any protein functional site remains difficult. To address this challenge, we develop a computational "anchor extension" methodology for targeting protein interfaces by extending a peptide chain around a non-canonical amino acid residue anchor. To test our approach using a well characterized model system, we design cyclic peptides that inhibit histone deacetylases 2 and 6 (HDAC2 and HDAC6) with enhanced potency compared to the original anchor (IC$_{50}$ values of 9.1 and 4.4 nM for the best binders compared to 5.4 and 0.6 μM for the anchor, respectively). The HDAC6 inhibitor is among the most potent reported so far. These results highlight the potential for de novo design of high-affinity protein-peptide interfaces, as well as the challenges that remain.

[1] University of Washington, Department of Biochemistry, Institute for Protein Design, Seattle, WA, USA. [2] Roy and Diana Vagelos Laboratories, Department of Chemistry, University of Pennsylvania, Philadelphia, PA, USA. [3] Molecular and Cellular Biology Ph.D. Program, University of Washington, Seattle, WA, USA. [4] Department of Structural Biology, Stanford University School of Medicine, Stanford, CA, USA. [5] Molecular Engineering Ph.D. Program, University of Washington, Seattle, WA, USA. [6] Present address: Knight Campus Center, University of Oregon, Eugene, OR, USA. [7] Present address: Systems Biology, Center for Computational Biology, Flatiron Institute, New York, NY, USA. [8] Present address: Key Laboratory of Structural Biology of Zhejiang Province, School of Life Sciences, Westlake University, Hangzhou, Zhejiang Province, China. [9] Present address: Lyell Immunopharma, Inc., Seattle, WA, USA. [10] Present address: Novo Nordisk A/S, Måløv, Denmark. [11] These authors contributed equally: Paris R. Watson, Timothy W. Craven, Xinting Li, Stephen Rettie. ✉email: dabaker@uw.edu

Peptides are emerging as a promising class of therapeutics with the potential to bind protein surfaces that are difficult to target using small molecules[1–8]. Cyclic peptides have been of particular interest due to their tunable rigidity, stability, and pharmacokinetics properties[4,9]. Library-based peptide discovery methods have been used to obtain molecules that bind to protein interfaces with high affinity[4,10,11], with considerable progress in terms of library size (currently up to $10^{14}$), and ability to incorporate a variety of different amino acids[12–16] and cyclization chemistries[17–19]. Due to limitations in synthesis, however, sampling the entirety of the chemical space is rarely possible and these libraries are limited to a subset of amino acids, often guided by the biochemical properties of the protein surface of interest.

Structure-based design of cyclic peptide binders has been more challenging. Most current peptide binder design methods take advantage of one or more co-crystal structures of the target protein with a protein binding partner, and generate binders by stabilizing or scaffolding the interacting structural elements[20–24], or mimicking[25] or enhancing the binding interface by amino acid substitions[26]. The requirement for a co-crystal structure with a binding partner limits the application of these methods because for many target proteins no such structure is available. In addition, most protein-protein interactions involve considerable buried surface area; the peptides can only span a portion of this surface and hence generally have diminished binding affinity compared to the original binding partner. Restricting to known binding partners also significantly decreases the range of targetable surfaces.

In this paper we present a general computational approach for de novo design of cyclic peptides that bind to a target protein surface with high affinity. The three-dimensional structure of the target surface is needed for this approach and can be derived from an experimentally determined or computationally predicted protein structure. This method takes advantage of a functional group from a molecule known to bind to the target surface of interest which serves as an *anchor*, around which a cyclic peptide is built using the generalized kinematic loop closure method in Rosetta software. We generate macrocyclic scaffolds that place this anchor in a binding-competent orientation and enhance its binding to the target by providing additional interactions introduced during computational design. We call this strategy *anchor extension*.

## Results

**Choice of target protein and anchor residue**. We chose Histone deacetylase (HDACs) as model targets to test our approach because of the wealth of structural data available for these enzymes, the simplicity of testing binding through enzymatic inhibition, and the relative ease of growing crystals of HDACs which facilitates structural characterization of new designs. We focused our design efforts on HDAC2. In addition to its therapeutic relevance[27,28], HDAC2 exemplifies two of the major challenges facing binder design: First, it has a relatively polar surface (Supplementary Fig. 1a), making it a good test system for designing binders for hydrated surfaces. Second, HDAC2 belongs to a protein superfamily with 11 members, many of which share high structural homology[29,30] (Fig. 1a, Supplementary Fig. 1b); thus, being able to selectively bind to only HDAC2 present a challenge for computational design of selectivity.

The HDACs are particularly well suited as paradigm systems for the development of our anchor-extension approach. Smaller peptides or peptide-like inhibitors, such as the marine depsipeptide Largazole (Fig. 1b), exhibit a range of affinities and selectivities against various HDAC isozymes. Many of these compounds were originally found in nature and in their unmodified forms bind with IC$_{50}$ values in the mid-nanomolar range or better, often to class I HDACs, with varying selectivities (Supplementary Table 1)[31,32]. We sought to determine whether computational methods can achieve similar or better inhibition than these natural products.

To design cyclic-peptide inhibitors of HDAC2, we chose a non-canonical amino acid, 2S-2-amino-7-sulfanylheptanoic acid (SHA), as the anchor. SHA can coordinate to the zinc ion in the active site of HDACs. The choice of SHA was inspired by natural product Largazole (Fig. 1b)[33], one of the most potent naturally occurring HDAC inhibitors (1.2 nM for HDAC1, ~3 nM for HDAC2 and HDAC3, and 4.6 nM for HDAC6)[34]. SHA alone can inhibit class I (HDACs 1,2,3, and 8) and class IIb (HDAC6) HDACs with low micro-molar to high nano-molar affinities (Supplementary Table 2). Different conformations of the SHA anchor were sampled in the HDAC2 pocket using molecular dynamics simulations and served as starting points for design.

**Docking pre-existing scaffolds**. Our first computational method (design method 1) started with docking structured cyclic-peptide scaffolds onto SHA and re-designing the residues in the interface to improve binding. These scaffolds were selected from two peptides with known structure in the Protein Data Bank (PDB): 3AVL-chain C and 3EOV-chain C, as well as a library of 200 previously generated computationally designed[35] structured cyclic-peptides of 7-10 residues length. These peptides were docked onto different conformations of SHA embedded in the HDAC2 active site by rigid body superposition. We selected docked poses without clashes between the macrocycle backbone and HDAC2, and then redesigned the peptide sidechains using Rosetta combinatorial sequence optimization to maximize predicted affinity for the HDAC2. Designs were ranked based on shape complementarity between the peptide and the protein pocket, calculated ΔΔG of binding, and number of contacts between peptide and protein (see Methods for more details). From a library of tens of thousands of designed peptides, 100 with the best interface metrics were selected for energy landscape characterization. Tens of thousands of conformers were generated for each of the 100 peptides, their energies were evaluated, and those designs for which the designed target structure had the lowest energy were selected for downstream analysis. From the previous pool, five peptides with the greatest in silico predicted affinities were tested for HDAC2 inhibition in vitro, the best of which (des1.1.0, Fig. 2a) demonstrated an IC$_{50}$ value of 289 nM (Fig. 2b, Supplementary Table 2).

The crystal structure of this design in complex with HDAC2 revealed two distinct binding modes and peptide conformations in the three protein chains of the asymmetric unit, suggesting flexibility of the peptide as well as presence of iso-energetic binding conformations (Supplementary Fig. 2a). In two of the protein chains of the asymmetric unit (chains B and C), the peptide conformation was very similar to the design model (Fig. 2c), but the observed binding mode to HDAC2 differed from that of the design model (Fig. 2d). This binding orientation interacts with several water molecules through backbone-mediated hydrogen bonds (Fig. 2e, and Supplementary Fig. 3a). Rosetta-guided mutational scanning successfully stabilized the binding mode observed in the crystal structure (confirmed by structures of variants Lys4→Glu and Pro3→hydroxy-Pro, Supplementary Fig. 2b and Supplementary Figs. 3b–c) but failed to significantly improve affinity or selectivity (Supplementary Tables 2 and 3).

**Stabilizing the binding orientation by including water molecules and an additional anchor**. Design method 1 highlighted the

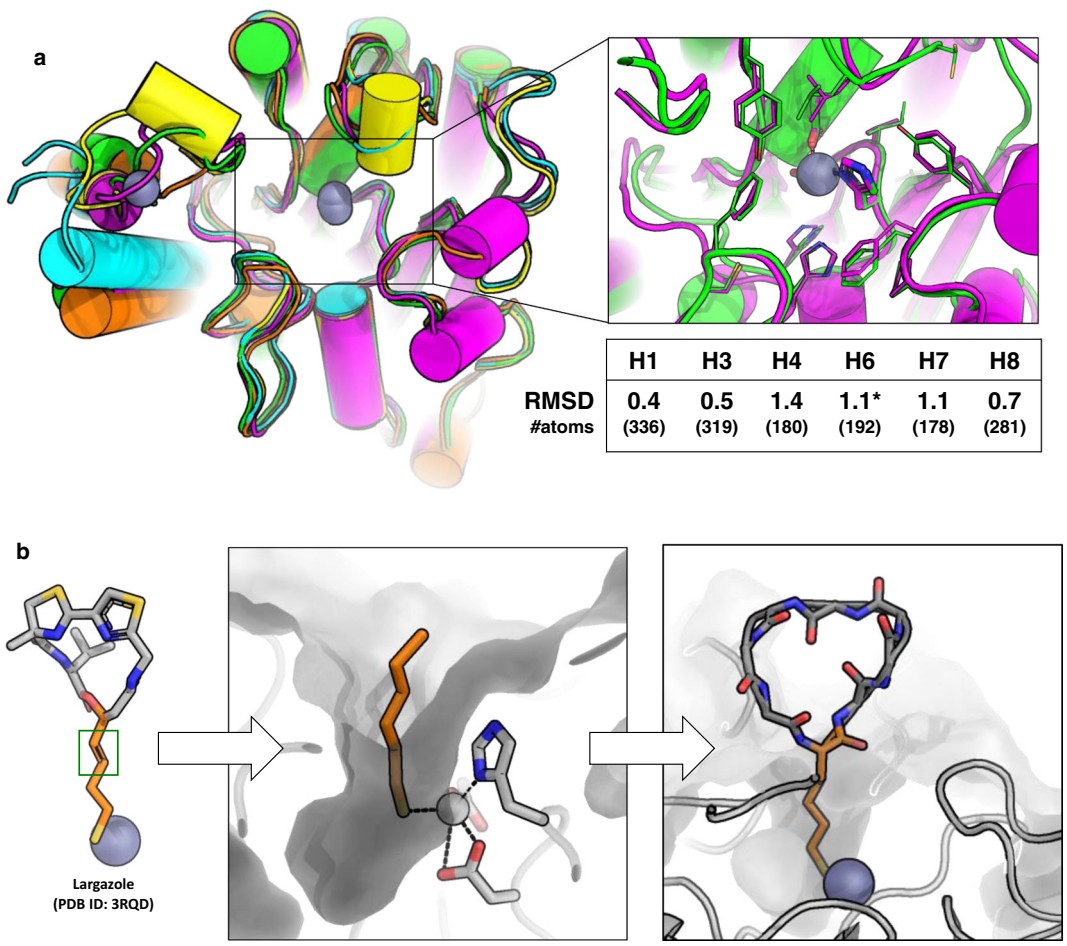

**Fig. 1 Anchor extension based design of macrocycles targeting HDAC2. a** Top left: Overlay of crystal structures of several HDACs showing the secondary structure elements adjacent to the active site (box on left) and their RMSD to HDAC2 (H1 = HDAC1, H3 = HDAC3, H4 = HDAC4, H6 = HDAC6, H7 = HDAC7, H8 = HDAC8). RMSD is calculated over the conserved secondary structure elements shown in the figure. Number of atoms in the aligned region is mentioned in the table. HDAC2 (green, PDB ID: 5IWG), HDAC4 (cyan, PDB ID: 2VQO), HDAC6 (magenta, PDB ID: 6R0K), HDAC7 (yellow, PDB ID: 3C0Z), HDAC8 (orange, PDB ID: 3SFF). All structures shown here are human variants except HDAC6 (shown by an asterisk), which is the homolog from *Danio rerio*. The inset on top right shows the overlay of the active site residues of HDAC2 (green) and HDAC6 (magenta). The residues coordinating the active site Zn (gray sphere) are shown in sticks. **b** Schematic representation of anchor-extension approach. The SHA (2S-2-amino-7-sulfanylheptanoic acid) anchor was inspired by the long tail of the HDAC-binding small molecule Largazole and modeled in HDAC2 pocket. A double bond in Largazole (green box) was replaced by a single bond in SHA to allow synthesis. Low-energy bound conformations were sampled using molecular dynamics simulations and served as starting points for designing new macrocycle binders.

limitation of using a small (~200) library of pre-designed scaffolds for obtaining high affinity binders: with this small a library, the best fit solutions are not likely to have high shape complementary and make optimal interactions with the target. To circumvent this problem, in design round 2, we built up macrocyclic scaffolds de novo inside the protein pocket to increase shape complementarity between the peptide and the HDAC2 pocket, and to make additional favorable interactions between the peptide backbone atoms and HDAC2. We also incorporated a Trp residue as an additional anchor that lays flat on a hydrophobic surface close to the active site (Fig. 3a), and mimics an aromatic ring observed in the same position in several small molecule inhibitors of HDACs (Supplementary Fig. 4a). Finally, since in the des1.1.0 crystal structure, the peptide interacts with two water molecules at the pocket of HDAC2, also present in a majority of HDAC2 crystal structures (Supplementary Fig. 4b), and displacing such tightly bound water is likely energetically unfavorable[36], we decided to explicitly include them in the design calculations and bias backbone sampling to start from

conformations that interact with structured waters at the HDAC2 interface (Fig. 3b).

The macrocycle chain was extended from the SHA-Trp dimer, and the conformational space of closed macrocycles harboring these two residues was explored using generalized kinematic loop closure[35]. Cyclic-peptide backbone and side-chains were optimized for increased predicted binding affinity and for stabilizing the binding competent conformation. As with method 1, out of tens of thousands of generated conformations, around 100 with best interface metrics were selected for conformational sampling analysis. Conformational sampling suggests that peptides designed with this method populate a wide range of conformations in the unbound state (Supplementary Fig. 5), thus we selected five peptides for experimental testing based on their in silico predicted affinities. The best tested peptide, des2.1.0 had an IC$_{50}$ value of 49.3 nM (Supplementary Table 2), a four-fold improvement over method 1. The other peptides from this method that were tested experimentally had IC$_{50}$ values in the low micro-molar to high nano-molar range. We hypothesize that the observed lower affinities are due in part to the flexibility of

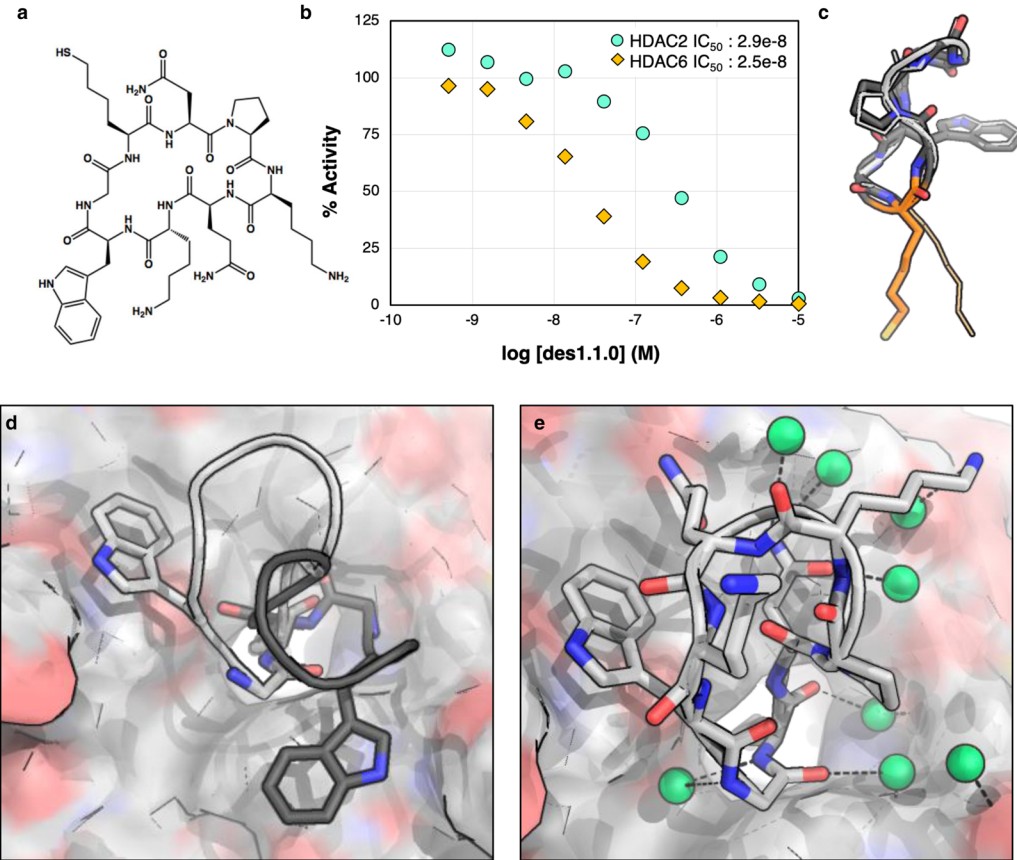

**Fig. 2 Crystal structure of des1.1.0 has the designed monomer structure but adopts a different binding orientation. a** Chemical structure of des1.1.0. **b** des1.1.0 inhibits HDAC2 with an $IC_{50}$ of 289 nM (Source Data are provided as a Source Data file). **c** Overlay of the designed peptide model des1.1.0 (dark gray) with the crystal structure (light gray). Some sidechains are removed for clarity. SHA anchor is shown in orange. **d** Overlay of des1.1.0 binding mode in original computational design (dark gray) and crystal structure (light gray) shows a clear rotation around SHA—$Zn^{2+}$ axis in the binding pocket. **e** Crystal structure of des1.1.0 (PDB ID: 6WHO) shows several water-mediated interactions at the interface. Water molecules are shown as green spheres.

these peptides, suggested by our computational conformational sampling (Supplementary Fig. 5), and the unfavorable loss of conformational entropy upon binding. This lower propensity to sample the designed competent conformation compared to designs from round 1 is potentially due to the constraints imposed by forcing the backbone to interact with water molecules.

To improve the affinity of des2.1.0, we mutated a D-Lys7 residue adjacent to a hydrophobic pocket in des2.1.0 to D-Met. The new des2.1.1 (Fig. 3c) had an improved $IC_{50}$ value of 16 nM for HDAC2 (Fig. 3d, Supplementary Table 2). The crystal structure of des2.1.1 matches the designed orientation for the two anchor residues, SHA and Trp, with the position of the water molecules retained (Fig. 3e, Supplementary Fig. 3d). While the electron density map for SHA and Trp are well resolved and match the designed model, the rest of the molecule appears to be in a conformation different from the solution structure obtained by NMR (Fig. 3f) and the designed model; however, the electron density is too weak or absent, and the thermal B-factors are too high, for those residues to enable a meaningful comparison. This flexibility is consistent with the observation of multiple states during conformational sampling (Fig. 3g).

**Anchor neighborhood sampling and identification of high potency binders**. While the above design methodologies result in macrocycles with modest affinities (methods 1 and 2) and correct binding orientation (method 2), these did not achieve selectivity over HDAC6 or $IC_{50}$ values in low nM range. Inspection of the

designed macrocycles in complex with HDAC2 suggested that most lacked sufficient hydrophobic contacts with the target interface to achieve high-affinity binding. The available scaffolds in method 1 and the backbone motif used in method 2 did not facilitate such interactions due to their shape and orientation, and the highly polar nature of HDAC2 pocket.

To overcome this challenge, we focused the anchor extension backbone generation towards making interactions with small hydrophobic patches around the active site (Supplementary Fig. 4c). To do so, we first sampled different possible conformations of SHA in the HDAC2 pocket to generate diversity in binding orientations. Next, the two residues directly adjacent to SHA were extensively sampled to find backbones that promote hydrophobic contacts to the protein surface using two parallel approaches, as described in the next two paragraphs.

In design method 3, we sampled the torsional space of these two residues using a grid-based search over all possible degrees of phi and psi (30° grids). After generating a library of backbones, the residue adjacent to SHA at each backbone torsion was mutated to all possible canonical amino acids (L chirality if φ < 0 and D chirality if φ > 0) and the shape complementarity and ΔΔG of binding were calculated for each mutation (Fig. 4a). The best scoring results from the grid-based sampling converged on a small number of residues and torsions, in particular for the residue after SHA (Supplementary Fig. 6). The solutions with best interface metrics were then extended into cyclic peptides by optimizing the torsions and amino acid sequence of the remaining residues as in method 2 through simultaneous

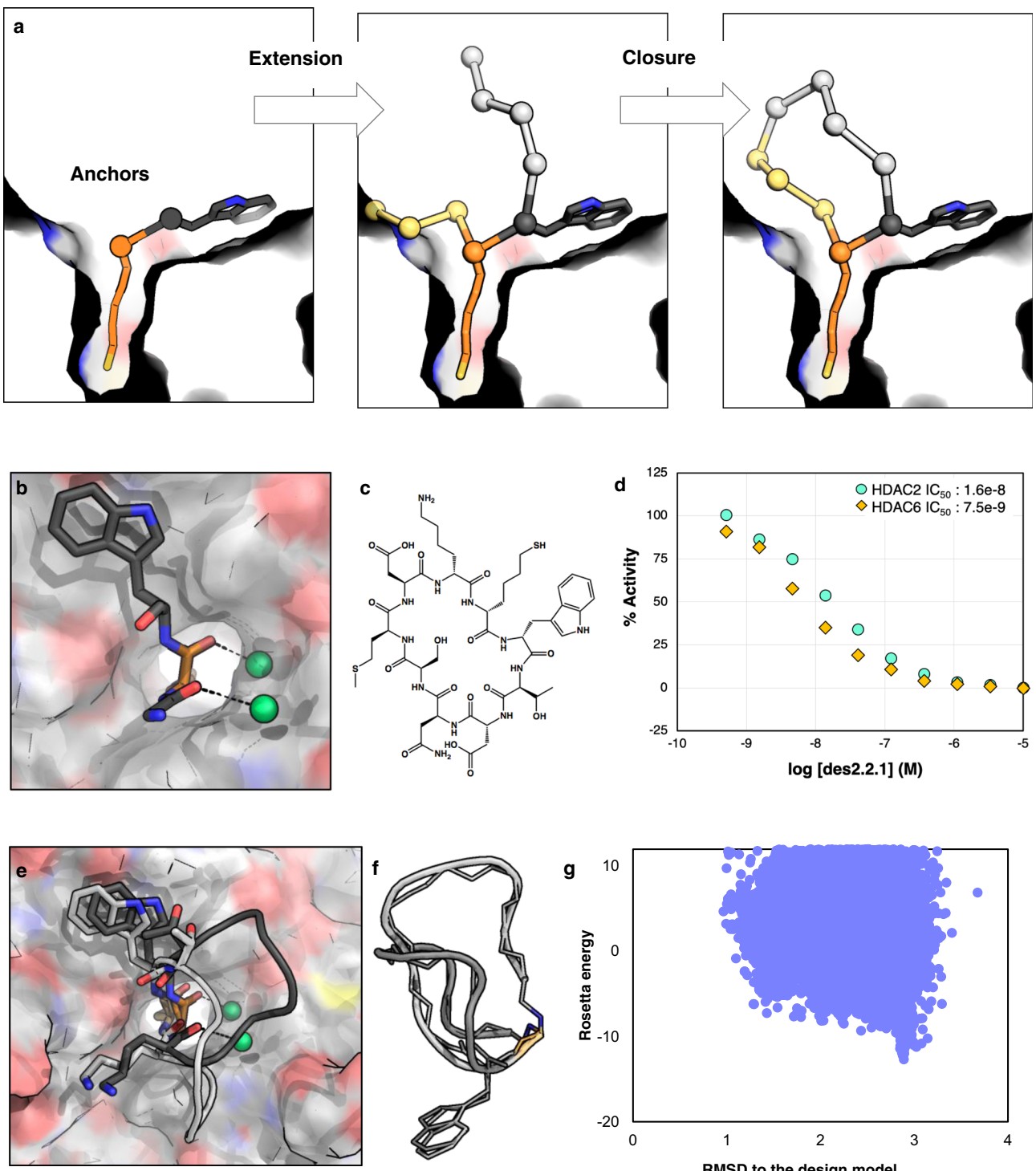

**Fig. 3 Design method 2 results in peptides that bind in predicted mode by taking advantage of an additional anchor and including waters. a** Schematic of design method 2: anchors are extended, and the new peptide chain is closed inside the protein pocket. **b** In this round of designs, a Trp residue was added as an anchor in addition to SHA, and the backbone orientation of SHA and the preceding residue were fixed to orient two structural waters at the interface. **c** Chemical structure of des2.1.1. **d** des2.1.1 has an $IC_{50}$ value of 16.3 nM against HDAC2 and slight preference for HDAC6 over HDAC2 (Source Data are provided as a Source Data file). **e** Crystal structure of des2.1.1 (PDB ID: 6WI3, light gray) shows a conformation consistent with the designed model for SHA, Trp, and waters at the interface; however, the rest of the peptide shows higher flexibility. The original model is shown in dark gray. **f** The binding-competent backbone conformation from the crystal structure (light gray) is different from the NMR structure of the peptide in solution (dark gray), suggesting a conformational change upon binding. Both of these structures differ from the design model. The sidechains, except for Trp are removed for clarity. **g** The flexibility of des2.1.1 is consistent with conformational sampling results, suggesting that this peptide can sample a number of different conformational states far from the designed model (Source Data are provided as a Source Data file).

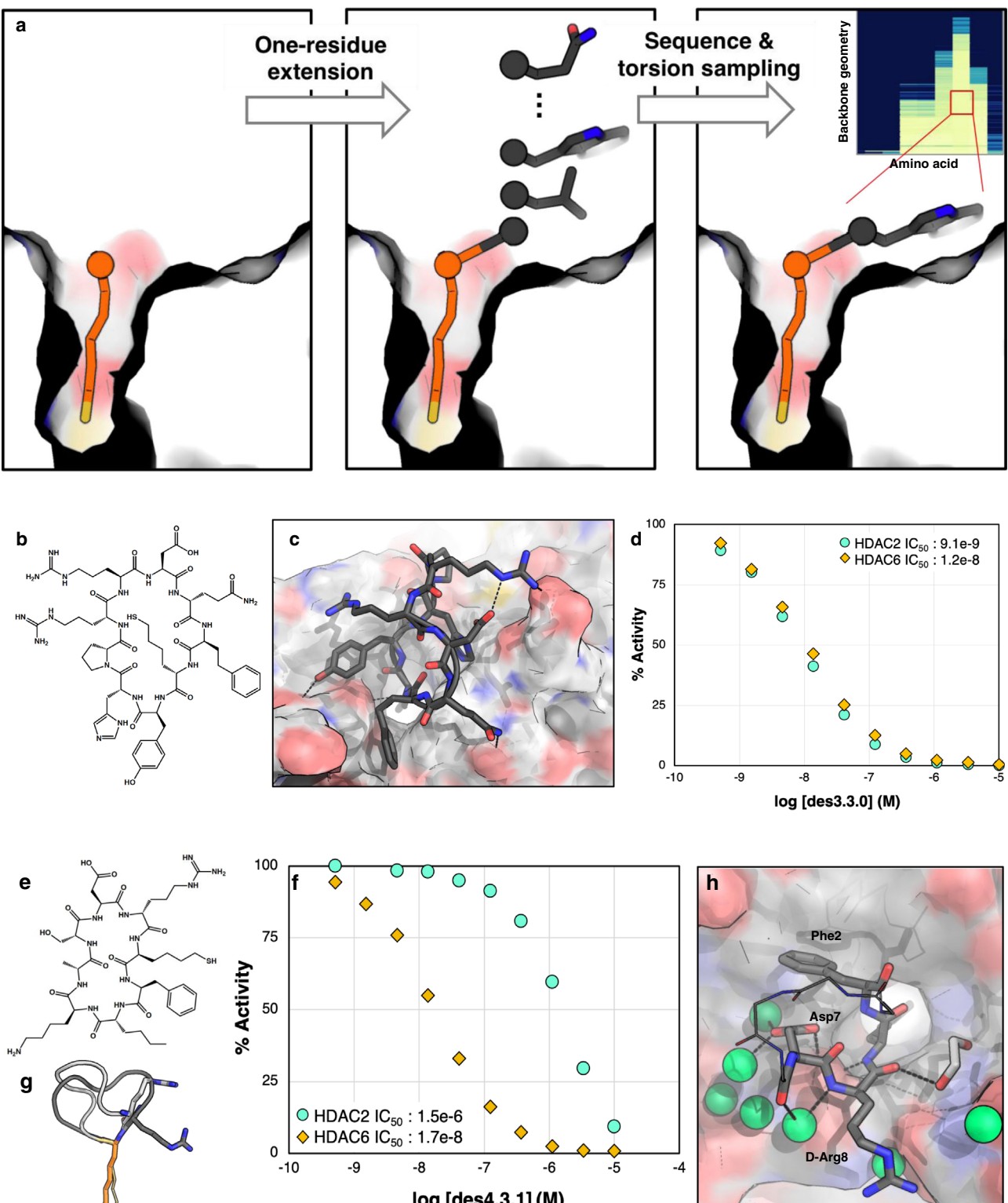

backbone sampling and Monte Carlo sequence design to favor binding interactions, including non-canonical amino acids. A list of noncanonical amino acids explored is provided in Supplementary Fig. 7. Except the SHA anchor, all positions were varied during the sidechain assignment due to the substantial backbone movement allowed in these approaches.

In design method 4, we biased backbone generation to favor more hydrophobic contacts by carrying out large-scale stochastic sampling of the two residues directly adjacent to the SHA anchor

followed by sequence optimization and energy minimization. In contrast to the grid-based approach in method 3, stochastic sampling joint with minimization and sequence optimization generates lower energy conformations with finer torsional diversity which can increase the likelihood of identifying favorable contacts. The 3mer peptides with best interface metrics were then extended to a final size of 7–9 residues and cyclic backbones were generated by sampling torsional space of these added residues. The peptide sequence was then optimized to

**Fig. 4 Design methods 3 and 4 generate cyclic peptides with higher shape complementarity to the binding pocket and better overall potencies. a** Schematic description of methods 3 and 4: the anchor is extended one residue before and after, and for each residue, backbone torsions are sampled. For each backbone geometry, the interface metrics are calculated for different amino acid substitutions for that residue. Inset: Example of $\Delta\Delta G$ distribution for a single residue position for different backbone geometries and amino acid choices. Backbone phi/psi distribution were sampled on 30° grids (each row is a different phi/psi bin), and the free energy of HDAC binding computed for different amino acid possibilities (y axis). $\Delta\Delta G$s are indicated in colors from light yellow (most favorable) to dark blue (most unfavorable). The best combinations of torsion and amino acid are then used for extension of the peptide sequence, closure, and design. **b** Chemical structure of des3.3.0 and **c** its computational model at the HDAC2 interface. **d** des3.3.0 has an $IC_{50}$ of 9.1 nM for HDAC2 and 12 nM for HDAC6 (Source Data are provided as a Source Data file). **e** Chemical structure of des4.3.1. **f** des4.3.1 inhibits HDAC6 with an $IC_{50}$ value of 17 nM, 88 times better than its potency for HDAC2 (Source Data are provided as a Source Data file). **g** Crystal structure of bound des4.3.1 (PDB ID: 6WSJ, light gray) is different from the designed model (dark gray). D-Arg8 (shown as sticks) adopts a negative phi torsion, a geometry more consistent with L-Arg. **h** Crystal structure of des4.3.1 (PDB ID: 6WSJ) complexed with HDAC6. The HDAC6 structure shows minimal change upon binding (RMSD of 0.17 Å for 302 Cα atoms compared with the apo-structure, PDB ID: 5EEM). The non-interacting residues are shown as lines and their side-chains are omitted for clarity. Water molecules are shown as green spheres.

improve shape complementarity and calculated $\Delta\Delta G$ of binding between peptide and protein.

The macrocycles generated using design methods 3 and 4 on average had better shape complementarity with the binding pocket compared to those from previous rounds (Supplementary Fig. 8). We sampled around 100,000 peptides from both methods. 100 designs with the best shape complementarity and $\Delta\Delta G$ of binding were selected for downstream conformational sampling analysis, as described before. 13 out of the 22 peptides tested had $IC_{50}$ values of less than 100 nM (Supplementary Table 4); the best binders had similar sequences (Supplementary Table 2) and were all from design round 3. Further mutations around this sequence to enhance interactions to HDAC2 or stabilize the designed binding conformation did not improve experimental binding affinity (Supplementary Table 5). The best binder, des3.3.0 (Fig. 4b, c) had an $IC_{50}$ value of 9.1 nM for HDAC2 (Fig. 4d, Supplementary Table 2). One of the peptides, des3.3.2, achieved 3-fold selectivity over HDAC6, an order of magnitude improvement in selectivity over the original SHA anchor (Supplementary Table 2). Designs from round 4 had moderate potency for HDAC2, similar to round 1 designs (Supplementary Table 2). However, they all showed selectivity for HDAC6.

Efforts to crystalize des3.3.0 were not successful. However, we obtained a crystal structure of des4.3.1 (Fig. 4e) with zebrafish (*Danio rerio*) HDAC6 at 1.7 Å resolution (Supplementary Fig. 3e). This peptide inhibits human HDAC6 with an $IC_{50}$ value of 17.1 nM (Fig. 4f), 88-fold more potently than HDAC2. The binding orientation of the peptide is similar to the design model, but the structure of the peptide is very different (Fig. 4g). In particular, D-Arg8 occupies the negative phi region of the Ramachandran map ($\varphi = 166°$ in the design and -46° in the crystal structure). If the D-Arg is replaced by L-Arg during conformational sampling, the lowest energy structures are very close to the crystal structure (Supplementary Fig. 9). The newly adopted conformation is complementary in shape to the contour of the active site (Fig. 4h, Supplementary Fig. 3e) and stabilized by several backbone-backbone hydrogen bonds and $n \rightarrow \pi^*$ interactions[37]. The peptide also forms both direct and water-mediated hydrogen bonding interactions with residues in the outer active site cleft of HDAC6 (Fig. 4h, Supplementary Fig. 3e). Notably Ser531 of HDAC6 donates a hydrogen bond to the carboxylate group of Asp7 and accepts a hydrogen bond from the backbone amide of SHA; Ser531 similarly accepts a hydrogen bond from the backbone amide of bound acetyl-lysine HDAC substrates[38]. Mutation of Asp7 to Ala did not result in a significant change in $IC_{50}$ for HDAC6 or HDAC2 (Supplementary Table 6), while mutation of D-Arg8 → D-Ser raised the $IC_{50}$ 10-fold, suggesting an essential role for this residue (Supplementary Table 6), possibly due to its complementarity to the negative electrostatic surface potential of HDAC6 in this region (Supplementary

Fig. 10). HDAC6 was not the target in the computational design calculations, and it is unclear whether the binding mode of the peptide to HDAC2 would be similar or different from that observed in the HDAC6 structure.

**Insights from crystal structures.** The cyclic-peptides in this study are among the largest active site-targeted ligands that have been co-crystallized with any HDAC to date. Previously studied HDAC-cyclic peptide complexes include HDAC8 complexes with Largazole and Trapoxin A, and the HDAC6 complex with HC Toxin, each a tetrapeptide that interacts with specificity determinants in the enzyme active site[33,38,39].

In all the crystal structures we obtained, the thiol functional group of SHA coordinates the catalytic $Zn^{2+}$ ion with a $Zn^{2+}$– S separation of 2.3 Å in a distorted tetrahedral geometry reminiscent of that observed for the inhibitor Largazole (Supplementary Fig. 3)[33]. Binding of these peptides is enhanced by additional interactions to HDAC. Some of these interactions are observed in the crystal structure of shorter tetrapeptides, but there are also additional interactions due to the larger size of our designed peptides. For example, Ser531 in the HDAC6 active site accepts a hydrogen bond from the backbone NH group of the zinc-bound epoxyketone residue of HC Toxin, just as Ser531 accepts a hydrogen bond from the backbone NH group of substrate acetyllysine[38]. Similarly, Ser531 accepts a hydrogen bond from the backbone NH group of the zinc-bound anchor residue SHA; but in addition it forms water-mediated hydrogen bond interactions with main chain or side chain atoms at the mouth of the active site (Supplementary Fig. 3).

The crystal structure of HDAC1 complexed with a linear heptapeptide has been reported at 3.3 Å resolution, but solvent molecules are not modeled at this low resolution[40]. Our high-resolution crystal structures of HDAC2 and HDAC6 complexed with the large designed peptide ligands provide detailed clues regarding how these enzymes can interact with their protein substrates through direct and water-mediated hydrogen bond interactions.

Out of 5 HDAC2:cyclic-peptide crystal structures obtained, three had the same binding orientation and engaged the same residues at the protein-peptide interface as designed. In the des1.1.0:HDAC2 co-crystal structure, the dominant binding mode in the crystal (chains B and C) has considerably better predicted affinity than the original design model with the same peptide structure (Supplementary Table 7, Fig. 5a). Thus, the lack of structure recapitulation in this case was due to insufficient sampling—our original design calculations were not extensive enough to identify this lower energy alternative state. We found that a large-scale parallel docking algorithm which allows movements of side-chains and backbone of the peptide and nearby protein residues can reproduce the overall peptide

behavior in the binding pocket for this design (Fig. 5b). While encouraging, the resolution of this docking method is not high enough to capture the exact binding orientation and is computationally very expensive; these are clear areas for improvement.

Discrepancies between designed and experimental binding mode have been observed in other peptide- and protein-binder design studies[41,42], and are likely in part responsible for the limited binding affinities obtained using current computational methods[42,43].

## Discussion

Our anchor extension protocol shows promise for the de novo design of cyclic peptides capable of binding with nanomolar affinity to a protein surface of choice. The iteratively improved computational design protocols yielded HDAC2 and HDAC6 binders with affinities on par with many of the most potent inhibitors[44,45]. Our calculations include all 20 amino acids, their chiral variants and approximately 20 non-canonical amino acids (Supplementary Fig. 7) (a total of >$10^{13}$ possibilities), and the

high affinity peptide ligands described here were obtained by testing fewer than 50 peptides in vitro. These results indicate the power of the computational methods in narrowing down the space of potential peptide sequences to test experimentally. The correct orientation of SHA anchor in all crystal structures obtained in this study and the enhanced affinity of nearly all our tested peptides compared to this anchor further validates our approach. Out of 39 tested peptides (Supplementary Table 4), 30 have IC$_{50}$ values better than 1 μM, and 17 have IC$_{50}$ values of 100 nM or better (Fig. 5c, Supplementary Tables 2 and 4). Although selectivity is an important area for future work, des4.2.0, which is one of the most potent HDAC6 inhibitors reported (small molecule- or peptide-based, Supplementary Tables 1 and 8), shows that this class of molecules can be selective.

Our results also highlight remaining challenges for accurate and robust computational design of cyclic-peptide binders. In our previous work focused on ordered macrocycle design[35], the design calculations were entirely focused on maximizing the accuracy and stability of the monomeric designed structures. The additional requirement of harboring a high-affinity binding site

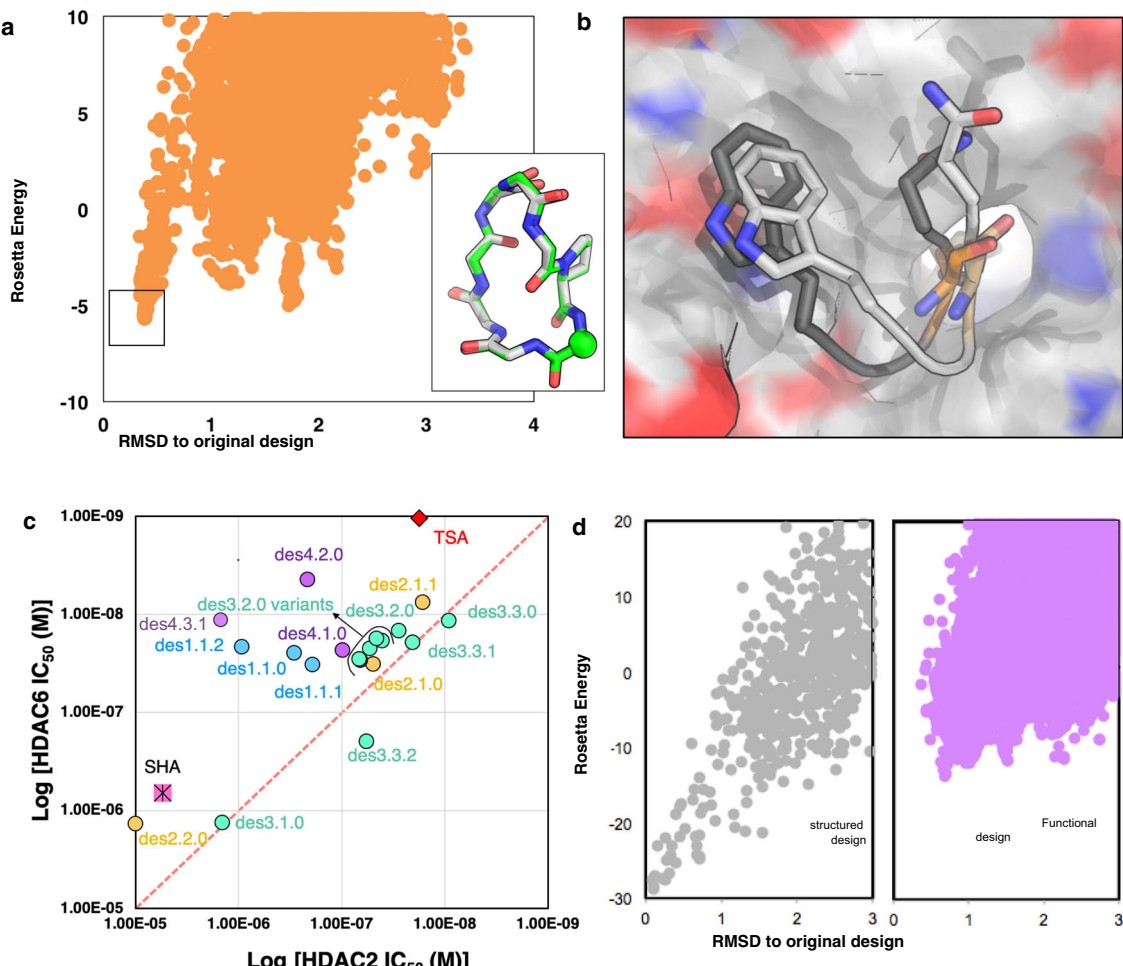

**Fig. 5 Improvements in scoring, structure sampling, and binding orientation sampling should increase design binding affinity and selectivity. a** Conformational sampling of des1.1.0 (Source Data are provided as a Source Data file). Inset: Overlay of the crystal structure (gray) and the best scoring model (green). The CA atom of the SHA anchor is shown as sphere for reference. **b** Overlay of predicted binding orientation from our large-scale parallel docking (dark gray) with crystal structure (light gray) of des1.1.0 shows that docking can accurately predict orientation of key residues at the interface. **c** Despite improvement of the IC$_{50}$ values over the original SHA anchor (pink square), most designs follow the same trend as SHA; binding slightly more tightly to HDAC6 over HDAC2. Different colors show results of designs from different methods (blue = method 1, orange = method 2, green = method 3, purple = method 4). TSA (Trichostatin A, red diamond), a pan-HDAC inhibitor, is shown as a control. **d** Comparison of computational conformational sampling for a structured (gray) and a functional (purple) macrocycle shows a much deeper energy gap for structured macrocycle compared to the functional macrocycle (Source Data are provided as a Source Data file).

constrains both the sequence and the structure of parts of the macrocycle and makes precise control over structure more challenging, and hence the designed structures were not always at the global free energy minima for the designed sequence (Fig. 5d); the constraints associated with incorporation of binding functionality reduce control over structure. This conformational flexibility also adds to the entropic cost of binding, a potential contributor to reduced affinity and lack of selectivity[46–49].

One solution for this problem is to sample a still larger number of possible macrocyclic conformations to identify those that satisfy both binding and structural criteria. Incorporating non-canonical amino acids with strong torsional preferences and cross-linking chemistries could help stabilize the designed binding conformation, compensating for the fact that a subset of residues must be optimized for binding rather than for folding to the desired structure. Faster and more accurate algorithms for computational sampling and energy evaluation to screen possible binding orientations will be important for confirming that the designed binding mode is indeed the lowest energy state. Taking into account the conformational ensemble of peptides (sampled using Rosetta[35] or MD-based methods[50,51]) when calculating binding rather than one single conformation could also be useful. Accurate energy calculations will likely have to consider structured water molecules[36,52–54] and flexibility of loops around the pocket[55–57] (reported previously for HDACs[58], also observed in our MD simulations, Supplementary Fig. 11). With improvements in both sampling methods and energy evaluation, and incorporation of a negative design strategy to increase within family selectivity, our anchor extension approach should enable the computational design of de novo peptides that can target "undruggable" surfaces with high affinity and selectivity.

## Methods
### Computational studies
*Preparation of proteins*. HDAC2 protein was obtained from PDB ID 5IWG ligated with BRD4884. Chain A of the protein was used as the HDAC2 template. All cofactors and water molecules were removed except the Zinc ion in the active site. The protein was then relaxed using the command below:

```
<path-to-Rosetta>/main/bin/relax.default.linuxgccrelease
-relax:constrain_relax_to_start_coords -relax:coord_
constrain_sidechains -relax:ramp_constraints false -score:
weights ref2015.wts -ex1 -ex2 -use_input_sc -flip_HNQ -no_optH
false -auto_setup_metals true -s <input_pdb>
```

For structure with water motif, used in design method 2, we added the flag -ignore_waters false.

*SHA parameterization and conformational sampling*. The SHA anchor by examining the x-ray crystal structure of Largazole bound to HDAC8 (PDB ID 3RQD). Since Fmoc-protected amino acid building-blocks with alkane side-chains of this sort were much more readily available from commercial suppliers than were the equivalent building blocks with the double bond intact, we decided to omit the double bond. To allow SHA to be modeled in Rosetta, we built the SHA anchor with L-chirality using the Avogadro software, appending N-acetyl and C-methylamide groups to emulate the context of a longer peptide. We energy-minimized the structure with the MMFF94 molecular mechanics force field, removed N- and C-capping groups, and converted the structure to Rosetta's params file format using the molfile_to_params.py script included with the software. We used the lysine main-chain potential for energetic calculations within Rosetta. Different conformations of SHA in HDAC2 pocket were sampled using molecular dynamics simulations with AMBER force field.

*Peptide backbone generation*. The first step in design is to generate the peptide in the pocket. For design method 1, a library of computational and few natural cyclic-peptides were docked onto different conformations of SHA anchor. The docking was performed using a python script that simply transforms the C, N, O, CA, and CB atoms of each peptide residue onto the corresponding atoms in SHA. This method is provided in supplementary data file 1, Method1 folder. The docked conformations were then relaxed using Rosetta FastRelax and those with minimal clashes were further carried out for design.

For design methods 2 the peptide chain was extended using PeptideStubMover and then closed conformations were sampled using GeneralizedKIC mover in Rosetta. The scripts are provided in supplementary data file 1.

For design methods 3 and 4, residues on two ends of SHA were sampled using a pyrosetta script provided in supplementary data file 1. The results were analyzed using python pandas and residues and orientations that had the best interface metrics (high shape complementarity and low ΔΔG) were selected for extension. A pyrosetta wrapper for GeneralizedKIC mover was developed in order to facilitate sampling different peptide sizes.

*Peptide design and selection*. Extended peptides were then designed using Rosetta's FastDesign mover[59]. The details of design were slightly different for different methods; details of each script is provided in the supplementary data file 1. The designed peptides were then filtered based on their total Rosetta score, shape complementarity, ΔΔG of binding, and contacts at interface. The analysis was performed using python pandas. For each design method, the designs in the top 1% of all these metrics were chosen for further analysis (approximately 100 designs in a total pool of 50,000-100,000). The threshold for each metric depended on the distribution of scores for each method. However, we employed a hard cut-off of shape complementarity >0.65 and calculated ΔΔG without repacking < −10 for all the designs.

Selected designed models were then visually inspected and those with minimum number of buried unsatisfied hydrogen bond donors and acceptors were selected for computational conformational sampling, or folding analysis. The folding analysis was performed using simple_cycpep_predict app in Rosetta as described before[35]. Below you may find a command to run such analysis:

```
<path_to_rosetta_binary>/simple_cycpep_predict.
default.linuxgccrelease \
  -nstruct 10000 \
  -cyclic_peptide:sequence_file seq.txt \
  -in:file:native native.pdb \
  -out:file:silent output.silent \
  -cyclic_peptide:genkic_closure_attempts 250 \
  -cyclic_peptide:genkic_min_solution_count 1 \
  -score:symmetric_gly_tables true \
  -cyclic_peptide:default_rama_sampling_table flat_symm_
pro_ramatable \
  -cyclic_peptide:use_rama_filter true \
  -cyclic_peptide:rama_cutoff 3.0 \
  -cyclic_peptide:min_genkic_hbonds 2 \
  -cyclic_peptide:min_final_hbonds 2 \
  -mute all \
  -unmute protocols.cyclic_peptide_predict
```

Where seq.txt is the 3-Letter amino acid sequence of the peptide separated by space and native.pdb is the designed conformation.

Peptides with better folding metrics (those that sampled near the binding conformation) were then selected based on score vs rmsd plot and moved forward for synthesis. Due to difficulties in finding peptides that sampled near the binding conformation in method 2, we selected peptides based on their interface metrics.

Parallel docking was performed using the pyrosetta.distributed suite and Dask parallel computing[60], using the environment settings provided in extended data. Details of running the docking and analysis are provided in a Jupyter notebook.

Visualization and figures were generated using the PyMOL Molecular Graphics System, Version 2.3.0 Schrödinger, LLC. The electrostatic surface potentials were generated using APBS PyMol plugin with default settings[61].

**Peptide synthesis and purification**. SHA was synthesized by Wuxi AppTech. All peptides were synthesized using standard Fmoc solid phase peptide synthesis protocols using a CEM Liberty Blue peptide synthesizer with microwave-assisted coupling and deprotection steps. Peptides containing a L-aspartate or L-glutamate were synthesized on a preloaded Fmoc-L-Asp(Wang resin LL)-ODmab or Fmoc-L-Glu(Wang resin LL)-ODmab resin, where the acidic side-chain of the amino acid is tethered to the resin, and were cyclized on-bead by a standard coupling reaction following deprotection of the C-terminal -ODmab protecting group with 2% (v/v) hydrazine monohydrate in dimethylformamide (DMF). All other peptides were synthesized with the C-terminus tethered to Cl-TCP(Cl) ProTide resin from CEM, cleaved from the resin with 1% (v/v) TFA in dichloromethane (DCM), and cyclized by a solution-phase coupling reaction prior to the final total deprotection. Crude peptides were purified based on mass via reverse phase HPLC using a Waters AutoPurify HPLC/MS system in line with a SQD2 mass spectrometer. Peptides were typically purified via a water (0.1% formic acid) and acetonitrile (0.085% formic acid) gradient at 2%/min on an XBridge Prep C18 10 um, 19 × 150 mm column. Masses and purities were assessed via electrospray ionization mass spectrometry during and subsequent to purification on an SQD2 mass spectrometer. Due to epimerization during cyclization, for some peptides two major peaks with correct mass was obtained. For such peptides, both peaks were collected and tested. LC/MS data for the peptides for which HDAC inhibition assays were performed are shown in Supplementary Fig. 12. Source Data are provided as Supplementary Data 2. Sequences of all peptides tested are provided in the Supplementary Data file 2.

**HDAC inhibition assay**. Initial IC$_{50}$ values were estimated using HDAC fluorogenic assay kits from BPS Biosciences (HDAC2, HDAC4, and HDAC6) following the protocol described in the kit. All the measures were performed in duplicates or

triplicates. These measurements were done either on full range of concentrations (1:10 dilution from 1 μM to 100 pM) or on two concentrations only (20 nM and 200 nM) as an estimate of the range. At this stage, the $IC_{50}$ value of the peptide was roughly estimated (reported in Supplementary Table 4) and on the promising hits, full $IC_{50}$ measurements for HDACs were performed by Reaction Biology Corporations as described below (Values reported in Supplementary Tables 2 and 4). Since no binding was observed to HDAC4, we did not perform full assay on HDAC4. Similarly, because no binding to HDACs 5,7,9,11 was observed for our SHA anchor, further binding analysis for these HDACs was not performed. Source Data are provided as a Source Data file.

The assay was performed using 50 μM fluorogenic peptide from p53 residues 379-382 [RHKK(Ac)AMC] in a base reaction buffer of 50 mM Tris-HCl at pH 8.0, 137 mM NaCl, 2.7 mM KCl, 1 mM $MgCl_2$, and freshly added 1 mg/ml BSA and 1% DMSO. Peptides were first dissolved in DMSO and then added to the reaction mixture with the enzyme using acoustic technique, enough to reach the desired concentration for that assay point. Peptide concentrations were calculated based on mass of lyophilized peptide and its molecular weight. The substrate was then added and incubated for 1 hour at 30 °C in a sealed container. After the incubation, developer was added to stop the reaction and develop color. Kinetic measurements were then performed for 20 min with Envision with 5 min intervals (Ex/Em = 360/460 nm). The endpoint after plateau was taken for $IC_{50}$ measurement. To calculate IC50, the % activity data are fit to the following equation in GraphPad Prism (Eq. 1).

$$Y = \text{Bottom} + \frac{(\text{Top} - \text{Bottom})}{1 + 10^{(\log(IC50) - X) * \text{HillSlope}}} \qquad (1)$$

where bottom is constrained to equal 0, and Top is constrained to be less than 120.

In cases where duplicate measures are used, average data is reported in the main text or Supplementary Tables.

### Structural studies

*NMR spectroscopy.* We used a Bruker (Billerica, MA) AVANCE III-800 NMR spectrometer that was equipped with a cryo-probehead to collect NMR spectra. Peptide concentrations of around 1.8 mM were used for collecting 1D and 2D $^1$H NMR spectra. The lyophilized peptides were dissolved in 90% $H_2O$:10% $D_2O$ solutions. To collect Total correlated spectroscopy (TOCSY) spectra, we used a mixing time of 120 ms applied with Bruker's mlevesgpph pulse sequence. A 250 ms mixing time using Bruker's noesyesgpph pulse sequence was used to collect 2D $^1$H−$^1$H nuclear Overhauser effect spectroscopy (NOESY) spectra. The TOCSY and NOESY peaks were assigned using the Sparky NMR package. NMR data are available in Supplementary Table 9 and Supplementary Fig. 13.

*Crystallography*

### Protein purification and crystallization

Protein of human HDAC2 was produced following the procedure described in Bressi et al.[62]. The full length, C-terminally His-tagged protein is expressed in insect cells and purified by affinity and size exclusion chromatography. After initial purification, a C-terminally truncated sample is generated by treatment with trypsin for 1 h at 25 °C. The addition of 1 mM PMSF is used to terminate the reaction, followed by a final purification by gel filtration chromatography and concentration to 12 mg/ml.

For crystallization of HDAC2, a 12 mg/ml protein sample was incubated with 1 mM tool compound (SHA) on ice for 1 hour. Experiments were conducted utilizing Takeda California's automated nanovolume crystallization platform, in which 50 nl of protein solution was mixed with an equal volume of reservoir solution containing 40% (v/v) PEG600 and 100 mM CHES (pH 9.5). The preformed HDAC2 crystals were used in subsequent soaking experiments by incubation for 48 hours in a stabilizing solution containing 10 mM of synthetic peptide. Crystals were subsequently harvested in a cryo-loop and frozen directly in liquid nitrogen.

HDAC6 catalytic domain 2 from *Danio rerio* (zebrafish; henceforth designated simply as "HDAC6"), as encoded in the $His_6$-MBP-TEV-HDAC6-pET28a(+) vector, was recombinantly expressed in *Escherichia coli* BL21(DE3) cells and purified as described[63]. The HDAC6–design4.3.1 complex was cocrystallized using the sitting drop vapor diffusion method at 4 °C. Briefly, a 100 nL drop of protein solution [10 mg/mL HDAC6, 5 0 mM 4-(2-hydroxyethyl)-1-piperazineethanesulfonic acid (HEPES) (pH 7.5), 100 mM KCl, 5% glycerol (v/v), 1 mM tris(2-carboxyethyl)phosphine (TCEP), and 2 mM design4.3.1 was combined with a 100 nL drop of precipitant solution [0.2 M ammonium chloride and 20% polyethylene glycol (PEG) 3350 (w/v)] and equilibrated against 80 μL of precipitant solution in the crystallization well reservoir. Rod-shaped crystals appeared within 4 days. Ethylene glycol (30% v/v) was added to the drop prior to crystal harvesting and flash cooling for data collection.

### Data collection

Data collection from crystals of HDAC2 complexes was performed with synchrotron radiation at both the Advanced Light Source (ALS) BL-5.0.3, and the Advanced Photon Source (APS), 21ID-F. All crystals belonged to space group $P2_12_12_1$ with cell dimensions closely related to the following, $a = 92.20$ Å, $b = 97.0$ Å and $c = 139.1$ Å, $\alpha = \beta = \gamma = 90°$. X-ray intensities and data reduction were evaluated by using the HKL2000 package[64].

HDAC6 X-ray diffraction data were collected at the Highly Automated Macromolecular Crystallography (AMX) beamline 17-ID-1 at the National Synchrotron Light Source II, Brookhaven National Laboratory (Upton, NY). Crystals of the HDAC6–design4.3.1 complex diffracted X-rays to 1.7 Å resolution. Data were indexed and integrated using iMosflm[65] and scaled using Aimless[66] in the CCP4 program suite[67].

### Refinement

Crystal structures of HDAC2 complexes were determined by molecular replacement using PDB 4LXZ as an initial search model with Phaser in the CCP4 Suite[67]. Model building and refinements were carried out using Phenix[68] and COOT[69] and models were manually corrected. After building the initial model, the peptide and solvent molecules were added. Phenix.refine was used for the final. Assessment and analysis of stereo-chemical parameters of the final model was done with Molprobity[70]. Data collection and refinement statistics are listed in Supplementary Table 10.

For the HDAC6–design4.3.1 complex, the initial electron density map was phased by molecular replacement using the program Phaser;[71] the atomic coordinates of unliganded HDAC6 (PDB 5EEM[38]) were used as a search model for rotation and translation function calculations. The interactive graphics program Coot[69] was used to build and manipulate the atomic model of the HDAC6–design4.3.1 complex, and refinement was performed using Phenix[68]. The atomic coordinates of design4.3.1 were built into the electron density map during the later stages of refinement. The final model was evaluated using MolProbity[72]. All data collection and refinement statistics are recorded in Supplementary Table 10.

*MD simulations.* Molecular Dynamics simulations of proteins HDAC10, HDAC3, HDAC6 and HDAC8 were performed using the Amber99SB-ILDN forcefield[73] with GROMACS 2016.1[74]. Each protein was solvated in a dodecahedron box of explicit TIP3P[75] water and neutralized with sodium ions. Each solvated and neutralized system was energy-minimized by steepest descent minimization. Equilibration of each system was performed for 1 ns under the NPT ensemble, where the pressure was coupled with the Berendsen barostat[76] to 1 atm, and the temperature was set to 310 K using the velocity-rescaling thermostat[77]. During the equilibration, position restraints (force of 1000 kJ mol-1 nm-1) were applied to heavy atoms. After equilibration, for each protein, 5 simulations of 100 ns were performed in the NVT ensemble, with periodic boundary conditions. A 10 Å cutoff was used for van der Waals and short-range electrostatic interactions. The Particle-Mesh Ewald (PME) summation method was used for long-range electrostatic interactions[78]. Verlet cut-off scheme was used[79]. Covalent bonds were constrained using the LINCS algorithm[80]. The integration time-step was 2 fs.

**Reporting summary.** Further information on research design is available in the Nature Research Reporting Summary linked to this article.

### Data availability

All the structures presented here are deposited in PDB with accession codes 6WHN, 6WHO, 6WHQ, 6WHZ, 6WI3, 6JSW. The raw data for HDAC inhibition assays (presented in Figs. 2–5 and Supplementary Table 2) are available as a supplementary data file. All relevant data are available from the authors upon request. Source data are provided with this paper.

### Code availability

Conformational sampling was done with the Rosetta `simple_cycpep_predict` application and peptide design was carried out with the `rosetta_scripts` application, both of which are included in the Rosetta software suite. The Rosetta software suite is available free of charge to academic users and can be downloaded from http://www.rosettacommons.org. Instructions and inputs for running these applications, and all other data and coding necessary to support the results and conclusion are provided in extended data files 1. Additionally, the code used for design and instructions on how to run can be found in Peptide_HDACBinders folder in our github repository (https://github.com/ParisaH-Lab/publications.git).

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

## Acknowledgements

This work was supported by ADRC grant (P50 AG005136, D.B.), NIH grant GM49758 (D.W.C.), Takeda Pharmaceuticals, NIH Ruth Kirschstein F32 award no. F32GM120791-02 (P.H.), Washington Research Foundation (P.H., T.W.C., and B.D.W), the Simons Foundation (V.K.M), NIH award R35GM122543 and Stanford Data Science Scholars program (F.P.), and EMBO ALTF 1605-2011 and Carlsbergfondet (P.G.). We thank the AMX beamline at the National Synchrotron Light Source II, a DOE Office of Science User Facility operated for the DOE Office of Science by Brookhaven National Laboratory under Contract No. DE-SC0012704. We also want to thank the Advanced Light Source (ALS) BL-5.0.3, and the Advanced Photon Source (APS), 21ID-F, for beamline use. We would like to acknowledge T. Asami, R. Skene, D. Cole, N. Hird and D. Verhelle at Takeda for collaborative support and the TCAL SB&B group for their great help with crystallography studies. We would like to thank Dr. L. Goldschmidt and P. Vecchiato for help with organization and set-up of computational resources. We would also like to thank Drs G. Bhardwaj, K. Deibler, A. Roy, S. Berger, P. Salveson, H. Haddox, F. Seeger, and H. Park for insightful scientific discussions, specifically Drs P. Salveson, K. Deibler, and S. Berger for their critical review of the manuscript. We also thank Prof. O. Furman from Hebrew University in Israel for scientific discussions. We also thank M. Murphy, Dr. J. Decarreau, L. Carter, and Drs N. Woodall and S. Berger for help with experiments that were not included in the paper.

## Author contributions

P.H. and D.B. developed the research idea, design strategies, and experimental approach. P.H., X.L., S.R., M.P. and J.G.W. performed peptide synthesis and purification. P.H. and P.L. generated designs and performed HDAC inhibition assays and analysis. F.P. performed MD simulation and analysis. P.H., V.K.M., A.S.F., B.D.W., P.G., and A.P.M. performed computational steps. T.W.C. ran NMR and P.H. and T.W.C. performed NMR assignment and structure analysis. P.R.W., D.W.C., and A.B. performed crystallization, data refinement and structure generation, and data deposition. L.J.S. helped with collaboration coordination and idea development. P.H. and D.B. wrote the main manuscript and all authors contributed to manuscript development, data generation and analysis.

## Competing interests

Rosetta software has been licensed to numerous not-for-profit and for-profit organizations. Rosetta Licensing is managed by UW CoMotion, and royalty proceeds are managed by the RosettaCommons. Under institutional participation agreements between the University of Washington, acting on behalf of the RosettaCommons, their respective institutions may be entitled to a portion of revenue received on licensing Rosetta software including programs described here. D.B. is an unpaid board member of Rosetta-Commons. V.K.M. is a co-founder of Menten AI, in which he holds equity. The remaining authors declare no competing interests.
