## [Peer Review File · Nature Communications]

Reviewers' Comments:

Reviewer #1:

Remarks to the Author:

The manuscript by Baker and colleagues describes a Rosetta-based workflow for the design of cyclic peptides for the binding of HDAC2. The workflow uses a molecular anchor as starting point, extends the scaffold in multiple iterations from there, and is in principle applicable to any protein with sufficient structural information. This is an important finding as computational de novo design of peptidic ligands is extremely challenging and a long standing goal. A large and convincing body of data is included (e.g. HDAC inhibition and crystal structures) and presented data illustrates the working principle nicely. Overall, I am impressed with the results and highly support publication of this manuscript after the following points have been addressed:

1) The introduction is very brief and previous work regarding the computational design/optimization of peptide ligands should be presented and put into context (pros and cons). In this respect, it is e.g. important to include a discussion on the consideration of water and ligand flexibility. Here is an incomplete list of related work:

- Rooklin J. Am. Chem. Soc 2017, 139, 15560
- Alogheli J. Chem. Inf. Model. 2017, 57, 190
- Antes Proteins: Struct., Funct., Genet. 2010, 78, 1084

2) The introduction should also briefly summarize the features of Rosetta that are important for this particular application (e.g. consideration of water is a very important aspect).

3) In the current form, the results section does not provide enough information to follow the workflow. Some information is hidden in the methods section, some I could not find at all. Overall, consider that the manuscript should be understandable for an scientist in a related field (but not necessary expert with Rosetta and peptide ligand design). E.g. What were the "pre-existing scaffolds" used in design round 1 and where did they come from? How big was the library in each step and how many members were selected for testing based on what criterion? ...

4) The results are described very much focused on structural aspects. The actual computational implementation and biochemical assays are only briefly discussed. More information should be provided here, e.g. how many structures were tested in each step, how many were tested, and preferably dedicating a sub-figure to the HDAC activity of the tested peptides. Also for key peptides, chemical structures should be presented (e.g. design3.2.1 and design4.3.1).

5) It would be helpful if the SI would contain a detailed version of Fig 5A having each of the dots labelled.

6) The discussion mentions "This conformational flexibility also adds to the entropic cost of binding, a potential contributor to reduced affinity and lack of selectivity." This is an aspect also briefly mentioned in the results part. The way in which differences in the flexibility of the unbound peptide influence complex stability should be discussed in more detail considering e.g. the following papers:

- Wallraven Chem. Sci., 2020, 11, 2269
- Kamenik J. Chem. Inf. Model., 2018, 58, 982
- Wagner ChemMedChem, 2017, 12, 1866
- Dagliyan Structure 2011, 19, 1837

7) When referencing to the SI, corresponding figure or table should be particularly mentioned

8) A figure depicting the used 20 non-natural amino acids should be added to the SI.

Reviewer #2:

Remarks to the Author:

The manuscript entitled: "Anchor extension: a structure-guided approach to design cyclic-peptides targeting histone deacetylases", submitted by Baker and coworkers, describes a new method for de novo design of HDAC inhibitors. This computational "anchor extension" method is tailored to fit histone deacetylase (HDAC) enzymes well, because the way HDACs interact with their substrates is harnessed in a thoughtful fashion. Thus, the fundamental idea behind the approach certainly deserves credit. Furthermore, the manuscript contains an impressive degree of structural validation of the produced hit peptides, by providing several x-ray co-crystal structures. Unfortunately, however, the developed peptides do not appear to provide any improvement beyond the state-of-the-art in the field of HDAC inhibitors (including cyclic peptide-based). The

authors state on page 2 that "being able to selectively bind to only HDAC2 and its close homologs HDAC1 and 3 (Fig. S1B) present a major design challenge". This is not an accurate statement, because several HDAC1–3 selective inhibitors exist, including cyclic peptides, which are actually usually more prone to inhibit HDAC1–3 than HDAC6. Examples of this include the approved drug romidepsin (FK-228), trapoxins, and apicidins.

Thus, the premise for why the developed peptides should be important is not valid in the opinion of this reviewer.

Furthermore, there is a lack of biochemical evaluation of the compounds, as full profiling against all HDACs (perhaps except for HDAC10, which is a polyamine deacetylase) should have been provided. Since this is not the case, I can only speculate as to whether any of the peptides would exhibit selectivity for one of the class I HDAC isoforms HDAC1–3, which would be a significant achievement. With the high sequence similarity between those three enzymes, however, this is unlikely. If significant isoform selectivity for HDACs 1, 2 OR 3 could be demonstrated and the below points were addressed, I would highly recommend reconsideration for publication in Nature Communications.

Revisions recommended before resubmission to a more specialized journal:

- 1) More detailed discussion of the state-of-the-art in peptide-based HDAC inhibitors.
- 2) Full profiling of inhibitors against the HDAC isoforms.
- 3) Proper characterisation of the peptides, including at least HPLC and HRMS, but preferably also copies of ¹H NMR spectra.
- 4) Docking of hit peptides to HDACs 1 and 3 for comparison of the interaction with HDAC2.
- 5) Evaluating in-cell activity of the hits, since these are semi-large peptide that may experience challenges with cell permeation.

Reviewer #3:

Remarks to the Author:

The use of peptides to target protein interfaces presents many opportunities and challenges. The facility of peptide chemistry and the ability to mimic or design compounds that target otherwise undruggable surfaces makes them an important class of molecules to develop. The lack of tertiary contacts however, makes them dynamic and difficult to stabilize in an active conformation. Cyclic peptides circumvent some of these challenges, but require modifications to established protein design methods to accommodate the cycle. In previous work, the Baker group successfully designed novel cyclic peptides by integrating analytic solutions for loop closure with the ROSETTA scoring and sampling tools. Here, cyclic peptides are designed to bind with high affinity and selectivity to a member of the HDAC class of proteins. The scientific work is of the highest quality and for the most part clearly described.

Minor comments.

It would be helpful to clarify the four Methods employed, where each appear to be sequential extensions or modifications to the previous approach. Description in the methods or a supplementary figure would be helpful.

Figure 4A was a little confusing. The central panel shows a number of amino acids layered - at first it looked like these were being connected to extend from the anchor, but probably it is meant to imply sequence sampling? The embedded figure on the right panel shows some quantitative information, but its not clear what the units for backbone geometry and amino acid would be.

More designs are mentioned in the supplement than sequences are provided. Sequences should be provided for all described designs.

A lot of values, particularly computed ones are listed with very high precisions going to the 4th or

5th decimal place. This should be amended to represent the significant digits.

Reviewer #1

The manuscript by Baker and colleagues describes a Rosetta-based workflow for the design of cyclic peptides for the binding of HDAC2. The workflow uses a molecular anchor as starting point, extends the scaffold in multiple iterations from there, and is in principle applicable to any protein with sufficient structural information. This is an important finding as computational de novo design of peptidic ligands is extremely challenging and a long standing goal. A large and convincing body of data is included (e.g. HDAC inhibition and crystal structures) and presented data illustrates the working principle nicely. Overall, I am impressed with the results and highly support publication of this manuscript after the following points have been addressed:

We appreciate reviewer #1's comment about our results. We also thank reviewer #1 for their insightful comments and suggestions. We tried the best we could to address all their concerns. Below is a line-by-line response and explanation:

1) Adding previous work on the computational design/optimization of peptides and their context to intro:

We have now revised our introduction to include a paragraph on this topic. Included in this paragraph is an explanation of previous work, its limitations, and several new citations including some of those suggested by the reviewer.

Revised/New text:

“Structure-based design of cyclic-peptides has been more challenging. Most current peptide binder design methods take advantage of one or more co-crystal structures of the target protein with a binding partner by stabilizing or scaffolding loops from the binding partner,^{20–24} enhancing the binding interface through mutations to canonical or noncanonical amino acids,²⁵ or mimicking the binding interface.²⁶ However, the requirement for a co-crystal structure limits the application of these methods because for many target proteins no such structure is available. In addition, most protein-protein interactions involve considerable buried surface area; the peptides can only span a portion of this surface and hence generally have diminished binding affinity compared to the original binding partner. Finally, restricting to known binding partners significantly decreases the range of targetable surfaces.”

New citations:

25. Rooklin, D. *et al.* Targeting Unoccupied Surfaces on Protein–Protein Interfaces. *J. Am. Chem. Soc.* **139**, 15560–15563 (2017).

26. Kadam, R. U. *et al.* Potent peptidic fusion inhibitors of influenza virus. *Science* **358**, 496–502 (2017).

2) The introduction should also briefly summarize the features of Rosetta that are important for this particular application (e.g. consideration of water is a very important aspect)

We have revised the introduction as well as the results section to emphasize more the specific features of Rosetta that were used. We also added more details to our methods section.

Revised/New text:

Last paragraph of introduction: “In this paper we present a general computational approach for de novo design of cyclic-peptides that bind to a target protein surface with high affinity. The three-

dimensional structure of the target surface is needed for this approach and can be derived from an experimentally determined or computationally predicted protein structure. This method takes advantage of a functional group in a molecule known to bind to the target surface of interest which serves as an *anchor*, around which a cyclic peptide is built using the generalized kinematic loop closure method in Rosetta software. We generate macrocyclic scaffolds that place this anchor in a binding-competent orientation and enhance its binding to the target by providing additional interactions introduced during computational design. We call this strategy *anchor extension*.”

Result sections: We revised the results section to include more details of how the design was performed, how we selected designs for experimental testing, and how many designs were computationally screened.

3) In the current form, the results section does not provide enough information to follow the workflow. Some information is hidden in the methods section, some I could not find at all. Overall, consider that the manuscript should be understandable for an scientist in a related field (but not necessary expert with Rosetta and peptide ligand design). E.g. What were the “pre-existing scaffolds” used in design round 1 and where did they come from? How big was the library in each step and how may members were selected for testing based on what criterion? ...

To address reviewer’s concern about ambiguity of the methods, we have highly revised the results section.

pre-existing scaffolds: “These scaffolds were selected from two peptides with known structure in the Protein Data Bank (PDB): 3AVL-chain C and 3EOV-chain, as well as a library of 200 previously generated computationally designed³⁵ structured cyclic-peptides of 7-10 residues length.”

Size of libraries and selection criteria:

Method 1: “Designs were ranked based on shape complementarity between the peptide and the protein pocket, calculated $\Delta\Delta G$ of binding, and number of contacts between peptide and protein (see Methods for more details). From a total library of tens of thousands of designed peptides, 100 with best interface metrics were selected for energy landscape characterization. Tens of thousands of conformers were generated for each of the 100 peptides, their energies were evaluated, and those designs for which the designed target structure had the lowest energy were selected for downstream analysis. From the previous pool, five peptides with the greatest *in silico* predicted affinities were tested for HDAC2 inhibition *in vitro*, the best of which (des1.1.0) demonstrated an IC₅₀ value of 289 nM (Fig.2A,B, Table S2).”

Method 2: “Cyclic-peptide backbone and side-chains were optimized for increased predicted binding affinity and for stabilizing the binding competent conformation. Similar to method 1, out of tens of thousands of generated conformations, around 100 with best interface metrics were selected for conformational sampling analysis. Conformational sampling suggests that peptides designed with this method, populate a wide range of conformations in the unbound state (Fig.S5), thus we selected five peptides for experimental testing based on their *in silico* predicted affinities.”

Method 3 and 4: “In design method 3, we sampled the torsional space of these two residues using a grid-based search over all possible degrees of phi and psi (30° grids). After generating a library of backbones, the residue adjacent to SHA at each backbone torsion was mutated to all possible canonical amino acids (L chirality if $\phi < 0$ and D chirality if $\phi > 0$) and the shape complementarity and $\Delta\Delta G$ of binding were calculated for each mutation (Fig. 4A). The best scoring results from the grid-based sampling converged on a small number of residues and torsions, in particular for the

residue after SHA (Fig.S6). The solutions with best interface metrics were then extended into cyclic peptides by optimizing the torsions and amino acid sequence of the remaining residues as in method 2 through simultaneous backbone sampling and Monte Carlo sequence design to favor binding interactions, including non-canonical amino acids. A list of noncanonical amino acids explored is provided in Fig.S7. Except the SHA anchor, all positions were varied due to the substantial backbone movement allowed in these approaches.

We also tested a different method (method 4) for sampling the backbone to favor more hydrophobic contacts. The two residues directly adjacent to the SHA anchor were sampled by large-scale stochastic sampling coupled with minimization and sequence optimization. In contrast to grid-based approach in method 3, stochastic sampling joint with minimization and sequence optimization generates lower energy conformations with finer torsional diversity which can increase the likelihood of identifying favorable contacts. The 3mer peptides with best interface metrics were then extended to a final size of 7-9 residues and cyclic backbones were generated by sampling torsional space of these added residues. The peptide sequence was then optimized to improve shape complementarity and calculated $\Delta\Delta G$ of binding between peptide and protein.

4) The results are described very much focused on structural aspects. The actual computational implementation and biochemical assays are only briefly discussed. More information should be provided here, e.g. how many structures were tested in each step, how many were tested, and preferably dedicating a sub-figure to the HDAC activity of the tested peptides. Also for key peptides, chemical structures should be presented (e.g. design3.2.1 and design4.3.1).

To address reviewer's comment, we have added more explanations about computational methods throughout the result section (see reply to question 3). We have also incorporated all chemical structures (both in main figures as well as in new Fig.S12) and added new figures and tables (new figures: Fig.2A-B, Fig3C,D,G, Fig.4B,D,E,F, Fig.S6-7, new tables: S1,3,5,6,8). We have also included the binding assay plots in our main figures. The raw data for all binding experiments is also included in Supplementary data file 2.

More explanation of computational methods: We have highly revised the results section to include the details asked (please see answer to question 3). We also included an additional SI figure (Fig. S6) with additional computational analysis. Finally, we added a section to results "Insight from crystal structures" to summarize the understanding gained from the structures and how it relates to computational metrics.

Peptide chemical structures: We have now added new figures for all chemical structures of peptides.

Inhibition assay: We have now added a figure for the inhibition assay results for all peptides either in the main text figures or in the SI material.

5) It would be helpful if the SI would contain a detailed version of Fig 5A having each of the dots labelled.

We have modified Fig.5C to include labels. The exact numbers are reported in Table S2.

6) The discussion mentions "This conformational flexibility also adds to the entropic cost of binding, a potential contributor to reduced affinity and lack of selectivity." This is an aspect also briefly mentioned in the results part. The way in which differences in the flexibility of the unbound peptide influence complex stability should be discussed in more detail considering e.g. the following papers:

We thank the reviewer for suggested articles and for pointing out to the limitation in our discussion. Based on reviewer's suggestion, we have modified the discussion and added relevant discussions.

Revised/New text:

Discussion: "Faster and more accurate algorithms for computational sampling and energy evaluation to screen possible binding orientations will be important for confirming that the designed binding mode is indeed the lowest energy state. Taking into account the conformational ensemble of peptides (sampled using Rosetta³⁵ or MD-based methods)^{47,48} when calculating binding rather than one single conformation could also be useful. Accurate energy calculations will likely have to consider structured water molecules^{36,49-51} and flexibility of loops around the pocket⁵²⁻⁵⁴ (reported previously for HDACs,⁵⁵ also observed in our MD simulations, Fig.S11). With improvements in both sampling methods and energy evaluation, our anchor extension approach should enable the computational design of *de novo* peptides that can target "undruggable" surfaces with high affinity and selectivity."

New citations:

46. Wallraven, K. *et al.* Adapting free energy perturbation simulations for large macrocyclic ligands: how to dissect contributions from direct binding and free ligand flexibility. *Chem. Sci.* **11**, 2269–2276 (2020).
47. Kamenik, A. S., Lessel, U., Fuchs, J. E., Fox, T. & Liedl, K. R. Peptidic Macrocycles - Conformational Sampling and Thermodynamic Characterization. *J. Chem. Inf. Model.* **58**, 982–992 (2018).
48. Yan, Y., Zhang, D. & Huang, S.-Y. Efficient conformational ensemble generation of protein-bound peptides. *J. Cheminformatics* **9**, 59 (2017).
52. Dagliyan, O., Proctor, E. A., D'Auria, K. M., Ding, F. & Dokholyan, N. V. Structural and dynamic determinants of protein-peptide recognition. *Struct. Lond. Engl.* **19**, 1837–1845 (2011).
53. Antes, I. DynaDock: A new molecular dynamics-based algorithm for protein-peptide docking including receptor flexibility. *Proteins Struct. Funct. Bioinforma.* **78**, 1084–1104 (2010).
54. Alogheli, H., Olanders, G., Schaal, W., Brandt, P. & Karlén, A. Docking of Macrocycles: Comparing Rigid and Flexible Docking in Glide. *J. Chem. Inf. Model.* **57**, 190–202 (2017).

7) When referencing to the SI, corresponding figure or table should be particularly mentioned

We thank the reviewer for catching this. We have gone through the manuscript and added the table/figure to parts that had been missing them.

8) A figure depicting the used 20 non-natural amino acids should be added to the SI.

We have now included Fig.S7 with chemical structures, names, and Rosetta 3 letter code of the noncanonical amino acids used in this study.

Reviewer #2

The manuscript entitled: "Anchor extension: a structure-guided approach to design cyclic-peptides targeting histone deacetylases", submitted by Baker and coworkers, describes a new method for de novo design of HDAC inhibitors. This computational "anchor extension" method is tailored to fit histone deacetylase (HDAC) enzymes well, because the way HDACs interact with their substrates is harnessed in a thoughtful fashion. Thus, the fundamental idea behind the approach certainly deserves credit. Furthermore, the manuscript contains an impressive degree of structural validation of the produced hit peptides, by providing several x-ray co-crystal structures.

We thank the reviewer for their comments, their appreciation of the approach as well as the structural studies. We have tried our best to address reviewer's concerns (below) by editing the manuscript and performing more experiments:

Unfortunately, however, the developed peptides do not appear to provide any improvement beyond the state-of-the-art in the field of HDAC inhibitors (including cyclic peptide-based). The authors state on page 2 that "being able to selectively bind to only HDAC2 and its close homologs HDAC1 and 3 (Fig. S1B) present a major design challenge". This is not an accurate statement, because several HDAC1–3 selective inhibitors exist, including cyclic peptides, which are actually usually more prone to inhibit HDAC1–3 than HDAC6. Examples of this include the approved drug romidepsin (FK-228), trapoxins, and apicidins. Thus, the premise for why the developed peptides should be important is not valid in the opinion of this reviewer.

We apologize for the ambiguity of the sentence. We have now changed the sentence to "thus, being able to selectively bind to only HDAC2 present a challenge for computational design of selectivity". Our point here is that in order to computationally design these binders from scratch one needs to overcome several known challenges in the field. We have modified the text accordingly to better explain this point.

Revised/New text:

Results (Choice of target): "The HDACs are particularly well suited as paradigm systems for the development of the anchor-extension approach for the design of de novo protein-peptide interfaces. Smaller peptides or peptide-like inhibitors, such as the marine depsipeptide Largazole (Figure 1), exhibit a range of affinities and selectivities against various HDAC isozymes. Many of these compounds were originally found in nature and in their unmodified forms bind with IC_{50} values in the mid-nanomolar range or better, often to class I HDACs, with varying selectivities (Table S1).^{31,32} We sought to determine whether computational methods can achieve similar or better inhibition than these natural products."

Results (Insight from crystal structures): "The cyclic-peptides in this study are the largest active site-targeted ligands that have been co-crystallized with any HDAC to date. Previously studied HDAC-cyclic peptide complexes include HDAC8 complexes with Largazole and Trapoxin A, and the HDAC6 complex with HC Toxin, each a tetrapeptide that interacts with specificity determinants in the enzyme active site.^{33,38,39} In all the crystal structures we obtained, the thiol functional group of SHA coordinates the catalytic Zn^{2+} ion with a $Zn^{2+}-S$ separation of 2.3 Å in a distorted tetrahedral geometry reminiscent of that observed for the inhibitor Largazole (Fig.S3).³³ Binding of these peptides is enhanced by additional interactions to HDAC. Some of these interactions are observed in the crystal structure of shorter tetrapeptides, but there are some novel interactions that are made possible due to the larger size of our designed peptides. For example, Ser531 in the HDAC6 active site accepts a hydrogen bond from the backbone NH group of the zinc-bound

epoxyketone residue of HC Toxin, just as Ser531 accepts a hydrogen bond from the backbone NH group of substrate acetyllysine.³⁸ 12/5/20 7:18:00 PM Similarly, Ser531 accepts a hydrogen bond from the backbone NH group of the zinc-bound anchor residue SHA; however, due to the large size of the cyclic peptide, it forms additional water-mediated hydrogen bond interactions with main chain or side chain atoms at the mouth of the active site (Fig.S3).”

New citations:

31. Kim, B. & Hong, J. An overview of naturally occurring histone deacetylase inhibitors. *Curr. Top. Med. Chem.* **14**, 2759–2782 (2015).
32. Salvador, L. A. & Luesch, H. Discovery and mechanism of natural products as modulators of histone acetylation. *Curr. Drug Targets* **13**, 1029–1047 (2012).
39. Porter, N. J. & Christianson, D. W. Binding of the Microbial Cyclic Tetrapeptide Trapoxin A to the Class I Histone Deacetylase HDAC8. *ACS Chem. Biol.* **12**, 2281–2286 (2017).

New data:

Additionally, we have now included a table (Table S1) describing the naturally occurring peptide-like inhibitors (or their variants) that are mentioned by the reviewer, highlighting that such inhibitors exist and providing a way to compare our results with existing compounds. We also included new Table S8 that compares our most potent HDAC6 inhibitors with other selective and potent HDAC6 inhibitors.

Furthermore, there is a lack of biochemical evaluation of the compounds, as full profiling against all HDACs (perhaps except for HDAC10, which is a polyamine deacetylase) should have been provided. Since this is not the case, I can only speculate as to whether any of the peptides would exhibit selectivity for one of the class I HDAC isoforms HDAC1–3, which would be a significant achievement. With the high sequence similarity between those three enzymes, however, this is unlikely. If significant isoform selectivity for HDACs 1, 2 OR 3 could be demonstrated and the below points were addressed, I would highly recommend reconsideration for publication in Nature Communications.

We apologize for poor data representation. We have initially performed full profiling for SHA and few of our original peptides. We have not detected binding to HDACs 4,5,7,9 for these peptides and the anchor, and thus stopped performing the assay on those HDACs.

We have now included profiling results on HDACs 1,2,3,6,8 and have provided results in Table S2. The raw data for these assays is reported in Supplementary data file 2.

While designed selectivity was not achieved, our results suggest that achieving selectivity is possible. We have obtained a peptide with modest selectivity for HDAC1 over HDAC2 (~12 folds), HDAC3 (~3 folds), and HDAC6 (~45 folds). We have also obtained a very potent HDAC6 inhibitor ($IC_{50} = 3.6$ nM) whose potency is on par with or better than currently existing HDAC6 inhibitors and exhibits good selectivities (see newly added Table S8).

In addition to their inhibitory properties, the peptides in this paper are the largest ligands ever co-crystallized with any HDACs. Since the native ligands of HDACs are proteins, we believe that these larger peptides can better mimic some of the features of biological interactions that can be missed in a smaller ligand. For example, we have observed many water-mediated hydrogen bonds between the ligand and the HDAC protein that is not observed in smaller tetrapeptides.

Revised/New text:

Results (Insight from crystal structures): “The cyclic-peptides in this study are the largest active site-targeted ligands that have been co-crystallized with any HDAC to date. Previously studied HDAC-cyclic peptide complexes include HDAC8 complexes with Largazole and Trapoxin A, and the HDAC6 complex with HC Toxin, each a tetrapeptide that interacts with specificity determinants in the enzyme active site.^{33,38,39} In all the crystal structures we obtained, the thiol functional group of SHA coordinates the catalytic Zn²⁺ ion with a Zn²⁺– S separation of 2.3 Å in a distorted tetrahedral geometry reminiscent of that observed for the inhibitor Largazole (Fig.S3).³³ Binding of these peptides is enhanced by additional interactions to HDAC. Some of these interactions are observed in the crystal structure of shorter tetrapeptides, but there are some novel interactions that are made possible due to the larger size of our designed peptides. For example, Ser531 in the HDAC6 active site accepts a hydrogen bond from the backbone NH group of the zinc-bound epoxyketone residue of HC Toxin, just as Ser531 accepts a hydrogen bond from the backbone NH group of substrate acetyllysine.³⁸ 12/5/20 7:18:00 PMSimilarly, Ser531 accepts a hydrogen bond from the backbone NH group of the zinc-bound anchor residue SHA; however, due to the large size of the cyclic peptide, it forms additional water-mediated hydrogen bond interactions with main chain or side chain atoms at the mouth of the active site (Fig.S3).”

Discussion: “Since our peptides are largest ligands that have been co-crystallized with any HDACs so far, they can provide a more accurate understanding of the biological interactions of HDACs and their protein partners. For example, the observation of largely water-mediated protein-peptide hydrogen bonds at the mouth of the active site and beyond suggests that protein-protein hydrogen bonds might be also mainly water-mediated in HDAC complexes with actual protein substrates.”

2) Full profiling of inhibitors against the HDAC isoforms.

We have initially performed full profiling for SHA and few of our original peptides. We have not detected binding to HDACs 4,5,7,9 for these peptides and the anchor, and thus stopped performing the assay on those HDACs.

We have now included profiling results on HDACs 1,2,3,6,8 and have provided results in Table S2. The raw data for these assays are reported in Supplementary data file 2.

3) Proper characterisation of the peptides, including at least HPLC and HRMS, but preferably also copies of 1H NMR spectra.

We apologize for not presenting the said data. We have now included HPLC and MS data of all the peptides presented in table S3 in new Fig.S12.

We have performed 1D NMR on a few peptides for which we had enough material in hand (data not presented). The results suggest that many can take more than one conformation in solution, making the result of this experiment less informative. Considering this result, the availability of LC/MS data that confirms the purification of correct peptide, and the limited access to experimental resources (peptide synthesis, NMR) due to covid, we decided to not run 1D NMR on other peptides. However, we have co-crystal structure of many of these peptides or their variants with HDACs, presented in the manuscript, that provide insight on the function of these peptides. We have now added a section named insights from crystal structures that highlight some of the insights we gained.

4) Docking of hit peptides to HDACs 1 and 3 for comparison of the interaction with HDAC2.

We thank the reviewer for this suggestion. We believe that the docking studies will be very interesting to have in hand to gain a better understanding of the interactions between peptides under study and HDACs, and we attempted running docking on our peptides.

Our peptides contain D-AAs, thus many available peptide docking methods cannot be used as they either use some means of fragment assembly from PDB fragments to sample the structure of peptides (for example flexpepdock or MDockPeP) and often only have parameters for L-AAs. Therefore, we limited our efforts to four methods that we knew could be applied to our peptides.

As a test, we docked des1.1.0 onto HDAC2. The reason to choose this peptide is that we obtained a co-crystal structure of this peptide in complex with HDAC2 with well-resolved peptide density at the binding pocket. Additionally, the structure of des1.1.0 matched the designed model, thus made the docking problem easier. Details of the results are provided below:

1. A Rosetta-based rigid body docking pipeline used for protein-protein docking: While this method did sample the experimental orientation, the predicted binding mode was the computational model. Additionally, this method does not sample the structure of the peptide, thus will fail for cases where the bound conformation of the peptide is different from the designed model.
2. Molecular dynamics simulations: We performed 50 ns MD simulations on des1.1.0:HDAC2 complex. We then clustered the results of our simulation. None of the clusters had a structure that resembled the experimental results.
3. Parallelized rotation and scoring: This is the newly developed method discussed in the last section of the results in the paper. Parallel docking could get the general binding orientation (rotation with respect to the SHA axis) correct, but it could not predict the correct tilt. We have tried this method for a number of other peptides reported in this paper and we've seen similar results. Additionally, this method still faces challenges in predicting the correct bound structure for more flexible peptides.
4. A newly developed ligand docking method called GALigandDock: Even when we include the knowledge of input conformation, GALigandDock could not predict the binding conformation with high confidence. In two separate repeats, only 3 models captured the experimental results out of 39 and 25 top ranking decoys. This number was reduced to 1 out of 43 and 36 when the input structure was disregarded.

This result suggests that the correct binding mode cannot be deduced with high confidence from the docking results, although in most cases it is sparsely sampled. We attribute this challenge in docking to flexibility of the peptide structure, flexibility of the target protein, ability of peptide to freely rotate along the SHA axis in HDAC pocket, and presence of water-mediated interactions.

Given that structural information about the bound conformation of our designed peptides or the correct binding orientation in HDAC3 or HDAC1 is not available, predicting the binding conformation using our current computational methods is not practical and developing a robust docking method is beyond the scope of the current manuscript.

5) Evaluating in-cell activity of the hits, since these are semi-large peptide that may experience challenges with cell permeation.

The scope of this paper is limited to providing a new methodology and a steppingstone towards the eventual goal of a robust computational pipeline to design selective inhibitors rather than generating therapeutically relevant HDAC binders. HDAC proteins were used just as a proof of concept due to the simplicity of assaying the binding and ability to obtain structural data. In-cell activity or pharmacokinetic studies are thus not within the scope of the current study. However, as the reviewer suggested, cell permeation is a clear challenge for peptide-based inhibitors and an important next step in the field, and in our future endeavors.

Reviewer #3

The use of peptides to target protein interfaces presents many opportunities and challenges. The facility of peptide chemistry and the ability to mimic or design compounds that target otherwise undruggable surfaces makes them an important class of molecules to develop. The lack of tertiary contacts however, makes them dynamic and difficult to stabilize in an active conformation. Cyclic peptides circumvent some of these challenges, but require modifications to established protein design methods to accommodate the cycle. In previous work, the Baker group successfully designed novel cyclic peptides by integrating analytic solutions for loop closure with the ROSETTA scoring and sampling tools. Here, cyclic peptides are designed to bind with high affinity and selectivity to a member of the HDAC class of proteins. The scientific work is of the highest quality and for the most part clearly described.

We thank the reviewer for their comments. We have tried to address their minor comments the best we could:

1. It would be helpful to clarify the four Methods employed, where each appear to be sequential extensions or modifications to the previous approach. Description in the methods or a supplementary figure would be helpful.

We have now modified the results section to make each method clearer. We have also modified our figures and tables (new figures: Fig.2A-B, Fig3C,D,G, Fig.4B,D,E,F, Fig.S6-7, new tables: S1,3,5,6,8) and elaborated more on the methods in the methods section. Some of the main text modifications are outlined below:

pre-existing scaffolds: “These scaffolds were selected from two peptides with known structure in the Protein Data Bank (PDB): 3AVL-chain C and 3EOV-chain, as well as a library of 200 previously generated computationally designed³⁵ structured cyclic-peptides of 7-10 residues length.”

Method 1: “Designs were ranked based on shape complementarity between the peptide and the protein pocket, calculated $\Delta\Delta G$ of binding, and number of contacts between peptide and protein (see Methods for more details). From a total library of tens of thousands of designed peptides, 100 with best interface metrics were selected for energy landscape characterization. Tens of thousands of conformers were generated for each of the 100 peptides, their energies were evaluated, and those designs for which the designed target structure had the lowest energy were selected for downstream analysis. From the previous pool, five peptides with the greatest *in silico* predicted affinities were tested for HDAC2 inhibition *in vitro*, the best of which (des1.1.0) demonstrated an IC_{50} value of 289 nM (Fig.2A,B, Table S2).”

Method 2: “Cyclic-peptide backbone and side-chains were optimized for increased predicted binding affinity and for stabilizing the binding competent conformation. Similar to method 1, out of tens of thousands of generated conformations, around 100 with best interface metrics were selected for conformational sampling analysis. Conformational sampling suggests that peptides designed with this method, populate a wide range of conformations in the unbound state (Fig.S5), thus we selected five peptides for experimental testing based on their *in silico* predicted affinities.”

Method 3 and 4: “In design method 3, we sampled the torsional space of these two residues using a grid-based search over all possible degrees of phi and psi (30° grids). After generating a library of backbones, the residue adjacent to SHA at each backbone torsion was mutated to all possible canonical amino acids (L chirality if $\phi < 0$ and D chirality if $\phi > 0$) and the shape complementarity and $\Delta\Delta G$ of binding were calculated for each mutation (Fig. 4A). The best scoring results from the grid-based sampling converged on a small number of residues and torsions, in particular for the

residue after SHA (Fig.S6). The solutions with best interface metrics were then extended into cyclic peptides by optimizing the torsions and amino acid sequence of the remaining residues as in method 2 through simultaneous backbone sampling and Monte Carlo sequence design to favor binding interactions, including non-canonical amino acids. A list of noncanonical amino acids explored is provided in Fig.S7. Except the SHA anchor, all positions were varied due to the substantial backbone movement allowed in these approaches.

We also tested a different method (method 4) for sampling the backbone to favor more hydrophobic contacts. The two residues around SHA anchor were sampled by large-scale stochastic sampling coupled with minimization and sequence optimization. In contrast to grid-based approach in method 3, stochastic sampling joint with minimization and sequence optimization generates lower energy conformations with finer torsional diversity which can increase the likelihood of identifying favorable contacts. The 3mer peptides with best interface metrics were then extended to a final size of 7-9 residues and cyclic backbones were generated by sampling torsional space of these added residues. The peptide sequence was then optimized to improve shape complementarity and calculated $\Delta\Delta G$ of binding.”

2. Figure 4A was a little confusing. The central panel shows a number of amino acids layered - at first it looked like these were being connected to extend from the anchor, but probably it is meant to imply sequence sampling? The embedded figure on the right panel shows some quantitative information, but its not clear what the units for backbone geometry and amino acid would be.

We have now added more description about figure 4A in the figure legend. We also have explained the method with more details in the results section to make the figure clearer. We also included a SI figure (Fig.S6) to show the results of this computational grid-based sampling.

Method 3 and 4 description: “In design method 3, we sampled the torsional space of these two residues using a grid-based search over all possible degrees of phi and psi (30° grids). After generating a library of backbones, the residue adjacent to SHA at each backbone torsion was mutated to all possible canonical amino acids (L chirality if $\varphi < 0$ and D chirality if $\varphi > 0$) and the shape complementarity and $\Delta\Delta G$ of binding were calculated for each mutation (Fig. 4A).”

Modified Fig.4A description: “Schematic description of methods 3-4. In these methods, the anchor is extended one residue before and after. For each residue, different torsional rotations are sampled and for each backbone geometry, the computational interface metrics are calculated for different amino acid substitutions for that residue. The inset figure shows a sample of the calculated computational $\Delta\Delta G$ of binding for different phi and psi geometries of amino acids. Each column is one amino acid and each row is one (φ, ψ) combination sampled in 30° grids. Light yellow colors have better computational $\Delta\Delta G$ whereas dark blue colors have unfavorable $\Delta\Delta G$ values (Rosetta energy units). The best combinations of torsion and amino acid are then used for extension of the peptide sequence, closure, and design.”

3. More designs are mentioned in the supplement than sequences are provided. Sequences should be provided for all described designs.

Sequences of all the peptides for which full HDAC profiling or initial HDAC inhibition has been performed are provided in Table S2 and Table S4, respectively. Additionally, HPLC and MS data for the peptides along with their sequence and chemical structure are reported in new Fig.S12.

4. A lot of values, particularly computed ones are listed with very high precisions going to the 4th or 5th decimal place. This should be amended to represent the significant digits.

We thank the reviewer for pointing out this issue. We have fixed the tables now.

Reviewers' Comments:

Reviewer #1:

Remarks to the Author:

In their revised version, the authors have adequately addressed the reviewer comments. I recommend publication of the manuscript.

Reviewer #2:

Remarks to the Author:

The authors have done an excellent job at revising their manuscript, addressing very well the concerns communicated by the reviewers.

There are still two issues that need attention before I can recommend publication of the work.

1) It is appreciated that the authors have toned down the importance of the developed peptides compared to the state-of-the-art in the literature. Now the introduction and discussion very nicely focus on the develop computational method and its potential. One addition to the new discussion, however, that may be a little exaggerated is the following sentence: "Since our peptides are largest ligands that have been co-crystalized with any HDACs so far, they can provide a more accurate understanding of the biological interactions of HDACs and their protein partners." While this is not untrue. Schwabe and coworkers have solved a co-crystal structure of HDAC1 with a linear heptapeptide, i.e., just one amino acid residue shorter (Nat. Commun. 7, 11262 (2016)). Likely because that peptide was linear and therefore more flexible, not all residues were resolved in the structure, but this seems to be a study that should be discussed in the present manuscript.

2) It is also highly appreciated that the authors now provide characterization of their synthesized peptides. However, the HPLC traces reveal that some of these peptides are substantially below purities that would normally be accepted. It is highly unorthodox to report and discuss biological activities for isolated compounds with purity below 95%. Therefore, several compounds must be either repurified or resynthesized to be included in the manuscript (in particular 3.3.1. and 4.3.1 are problematic as judged by the naked eye, but others may also be problematic when integrated). Alternatively, the authors should revise their manuscript to not include the peptides that are not sufficiently pure.

Reviewer #3:

Remarks to the Author:

This revision clearly describes the development and evolution of an 'anchor extension' protocol for designing high-affinity and specific cyclic peptide inhibitors of enzyme active sites. Targeting HDACs with peptides is an appropriate choice as the active sites are largely polar, with isozyme specificity a significant challenge - one which peptides can address. The presentation of the computational protocol is easy to follow in the revision and the discussion of its limitations due to computational expense is important.

Whether these compounds are leads for actual inhibitors is less important than the development of the approach itself. Although a fair amount of expert intervention and optimization of the protocol is required to achieve nanomolar binders with target specificity, each generation of design methods reflects improvements based on a physical understanding of the challenges, which enhances the impact and the generality of this approach.

Figure 5C presents the optimization of specificity in a quantitative way. This is achieved primarily through positive design - optimization of affinity and structural homogeneity for the peptide against its target. In the discussion of computational challenges and future directions, it would be worthwhile mentioning the role of negative design in maximizing the energy gap between HDAC2 and 6, how this would effect the computational expense of design and how gap and affinity would be integrated in design selection.

There is a typo in the heading Data and Cod(e) availability.

Excellent work and a valuable contribution to the design field.

Reviewed by Vik Nanda

Reviewer #1

In their revised version, the authors have adequately addressed the reviewer comments. I recommend publication of the manuscript.

We thank reviewer #1 for their insightful comments and suggestions and for recommending our manuscript for publication.

Reviewer #2

The authors have done an excellent job at revising their manuscript, addressing very well the concerns communicated by the reviewers.

There are still two issues that need attention before I can recommend publication of the work.

We thank reviewer #2 for their comments and we tried to address their concerns the best we could, detailed below:

1) It is appreciated that the authors have toned down the importance of the developed peptides compared to the state-of-the-art in the literature. Now the introduction and discussion very nicely focus on the develop computational method and its potential. One addition to the new discussion, however, that may be a little exaggerated is the following sentence: "Since our peptides are largest ligands that have been co-crystalized with any HDACs so far, they can provide a more accurate understanding of the biological interactions of HDACs and their protein partners."

While this is not untrue. Schwabe and coworkers have solved a co-crystal structure of HDAC1 with a linear heptapeptide, i.e., just one amino acid residue shorter (Nat. Commun. 7, 11262 (2016)). Likely because that peptide was linear and therefore more flexible, not all residues were resolved in the structure, but this seems to be a study that should be discussed in the present manuscript.

We thank the reviewer for bringing this point to our attention. We have now updated the manuscript by removing the the above mentioned sentence form the discussion. Instead, we added the sentences below to our "Insight from Crystal Structure" section, which includes the work by Schwabe and coworkers as a new citation (**citation 40**):

"High-resolution crystal structures of HDAC2 and HDAC6 complexed with such large peptide ligands provide detailed clues regarding how these enzymes can interact with their protein substrates through direct and water-mediated hydrogen bond interactions. Of note, the crystal structure of HDAC1 complexed with a linear heptapeptide has been reported at 3.3 Å resolution, but solvent molecules are not modeled at this low resolution.⁴⁰ Accordingly, the possible role of water-mediated interactions in peptide binding to the HDAC1 active site cannot be compared to HDAC2 or HDAC6. Even so, it is clear that the protein landscape surrounding each HDAC active site readily accommodates both cyclic and linear peptides."

2) It is also highly appreciated that the authors now provide characterization of their synthesized peptides. However, the HPLC traces reveal that some of these peptides are substantially below purities that would normally be accepted. It is highly unorthodox to report and discuss biological activities for isolated compounds with purity below 95%. Therefore, several compounds must be

either repurified or resynthesized to be included in the manuscript (in particular 3.3.1. and 4.3.1 are problematic as judged by the naked eye, but others may also be problematic when integrated). Alternatively, the authors should revise their manuscript to not include the peptides that are not sufficiently pure.

We really appreciate the reviewer for bringing to our attention the impurities in LC traces provided. We have updated the data now and we clearly labeled table S2 to note where impurity was observed and could not be removed.

In particular, for peptide 3.3.1, we obtained two peaks that interconverted to each other readily to reach an equilibrium of 1:1. This peptide is not mentioned in the main text and the presence of these two peaks is clearly stated in table S2. Peptide 4.3.1 was purified to high purity and all the data regarding this peptide are updated. We have also updated Tables S1 and S8 accordingly to indicate des4.2.0 as current best HDAC6 binder. Additionally, we have updated the text to mention “modest” selectivity for this peptide instead of “good”.

Reviewer #3

This revision clearly describes the development and evolution of an 'anchor extension' protocol for designing high-affinity and specific cyclic peptide inhibitors of enzyme active sites. Targeting HDACs with peptides is an appropriate choice as the active sites are largely polar, with isozyme specificity a significant challenge - one which peptides can address. The presentation of the computational protocol is easy to follow in the revision and the discussion of its limitations due to computational expense is important.

Whether these compounds are leads for actual inhibitors is less important than the development of the approach itself. Although a fair amount of expert intervention and optimization of the protocol is required to achieve nanomolar binders with target specificity, each generation of design methods reflects improvements based on a physical understanding of the challenges, which enhances the impact and the generality of this approach.

We thank reviewer #3 for their comments and summary of our work.

Figure 5C presents the optimization of specificity in a quantitative way. This is achieved primarily through positive design - optimization of affinity and structural homogeneity for the peptide against its target. In the discussion of computational challenges and future directions, it would be worthwhile mentioning the role of negative design in maximizing the energy gap between HDAC2 and 6, how this would effect the computational expense of design and how gap and affinity would be integrated in design selection.

As the reviewer correctly pointed out, negative design is a very important next step for achieving selective binders. While the detailed discussion of negative design strategies and how they will be implemented is beyond the scopes of the current manuscript, we have now added a mention of the negative design as a future strategy for obtaining selective binders (highlighted text below)

“Accurate energy calculations will likely have to consider structured water molecules^{36,52–54} and flexibility of loops around the pocket^{55–57} (reported previously for HDACs,⁵⁸ also observed in our MD simulations, Fig.S11). With improvements in both sampling methods and energy evaluation, and incorporation of a negative design strategy to increase within family selectivity, our anchor

extension approach should enable the computational design of *de novo* peptides that can target “undruggable” surfaces with high affinity and selectivity.”

There is a typo in the heading Data and Cod(e) availability.

We have fixed the typo.

Excellent work and a valuable contribution to the design field.

We thank the reviewer for their comment.

Reviewers' Comments:

Reviewer #2:

Remarks to the Author:

The authors have now addressed all remaining concerns and I recommend acceptance for publication.

Christian A. Olsen